# ALL SMILES VARIATIONAL AUTOENCODER FOR MOLECULAR PROPERTY PREDICTION AND OPTIMIZATION

## ABSTRACT

Variational autoencoders (VAEs) defined over SMILES string and graph-based representations of molecules promise to improve the optimization of molecular properties, thereby revolutionizing the pharmaceuticals and materials industries. However, these VAEs are hindered by the non-unique nature of SMILES strings and the computational cost of graph convolutions. To efficiently pass messages along all paths through the molecular graph, we encode multiple SMILES strings of a single molecule using a set of stacked recurrent neural networks, harmonizing hidden representations of each atom between SMILES representations, and use attentional pooling to build a final fixed-length latent representation. By then decoding to a disjoint set of SMILES strings of the molecule, our All SMILES VAE learns an almost bijective mapping between molecules and latent representations near the high-probability-mass subspace of the prior. Our SMILES-derived but molecule-based latent representations significantly surpass the state-of-the-art in a variety of fully- and semi-supervised property regression and molecular property optimization tasks.

## 1 INTRODUCTION

The design of new pharmaceuticals, OLED materials, and photovoltaics all require optimization within the space of molecules (Pyzer-Knapp et al., 2015). While well-known algorithms ranging from gradient descent to the simplex method facilitate efficient optimization, they generally assume a continuous search space and a smooth objective function. In contrast, the space of molecules is discrete and sparse. Molecules correspond to graphs, with each node labeled by one of ninety-eight naturally occurring atoms, and each edge labeled as a single, double, or triple bond. Even within this discrete space, almost all possible combinations of atoms and bonds do not form chemically stable molecules, and so must be excluded from the optimization domain, yet there remain as many as $10^{60}$ small molecules to consider (Bohacek et al., 1996). Moreover, properties of interest are often sensitive to even small changes to the molecule (Stumpfe & Bajorath, 2012), so their optimization is intrinsically difficult.

Efficient, gradient-based optimization can be performed over the space of molecules given a map between a continuous space, such as $\mathbb{R}^n$ or the $n$-sphere, and the space of molecules and their properties (Sanchez-Lengeling & Aspuru-Guzik, 2018). Initial approaches of this form trained a variational autoencoder (VAE) (Kingma & Welling, 2013; Rezende et al., 2014) on SMILES string representations of molecules (Weininger, 1988) to learn a decoder mapping from a Gaussian prior to the space of SMILES strings (Gómez-Bombarelli et al., 2018). A sparse Gaussian process on molecular properties then facilitates Bayesian optimization of molecular properties within the latent space (Dai et al., 2018; Gómez-Bombarelli et al., 2018; Kusner et al., 2017; Samanta et al., 2018), or a neural network regressor from the latent space to molecular properties can be used to perform gradient descent on molecular properties with respect to the latent space (Aumentado-Armstrong, 2018; Jin et al., 2018; Liu et al., 2018; Mueller et al., 2017). Alternatively, semi-supervised VAEs condition the decoder on the molecular properties (Kang & Cho, 2018; Lim et al., 2018), so the desired properties can be specified directly. Recurrent neural networks have also been trained to model SMILES strings directly, and tuned with transfer learning, without an explicit latent space or encoder (Gupta et al., 2018; Segler et al., 2017).

SMILES, the simplified molecular-input line-entry system, defines a character string representation of a molecule by performing a depth-first pre-order traversal of a spanning tree of the molecular graph, emitting characters for each atom, bond, tree-traversal decision, and broken cycle (Weininger, 1988). The resulting character string corresponds to a flattening of a spanning tree of the molecular graph, as shown in Figure 1. The SMILES grammar is restrictive, and most strings over the appropriate character set do not correspond to well-defined molecules. Rather than require the VAE decoder to explicitly learn this grammar, context-free grammars (Kusner et al., 2017), and attribute grammars (Dai et al., 2018) have been used to constrain the decoder, increasing the percentage of valid SMILES strings produced by the generative model. Invalid SMILES strings and violations of simple chemical rules can be avoided entirely by operating on the space of molecular graphs, either directly (De Cao & Kipf, 2018; Ma et al., 2018; Li et al., 2018; Liu et al., 2018; Simonovsky & Komodakis, 2018) or via junction trees (Jin et al., 2018).

Every molecule is represented by many well-formed SMILES strings, corresponding to all depth-first traversals of every spanning tree of the molecular graph. The distance between different SMILES strings of the same molecule can be much greater than that between SMILES strings from radically dissimilar molecules (Jin et al., 2018), as shown in Figure 8 of Appendix A. A generative model of individual SMILES strings will tend to reflect this geometry, complicating the mapping from latent space to molecular properties and creating unnecessary local optima for property optimization (Vinyals et al., 2015). To address this difficulty, sequence-to-sequence transcoders (Sutskever et al., 2014) have been trained to map between different SMILES strings of a single molecule (Bjerrum, 2017; Bjerrum & Sattarov, 2018; Winter et al., 2019b;a).

Reinforcement learning, often combined with adversarial methods, has been used to train progressive molecule growth strategies (Guimaraes et al., 2017; Jaques et al., 2017; Olivecrona et al., 2017; Putin et al., 2018; You et al., 2018; Zhou et al., 2018). While these approaches have achieved state-of-the-art optimization of simple molecular properties that can be evaluated quickly *in silico*, critic-free techniques generally depend upon property values of algorithm-generated molecules (but see (De Cao & Kipf, 2018; Popova et al., 2018)), and so scale poorly to real-world properties requiring time-consuming wet-lab experiments.

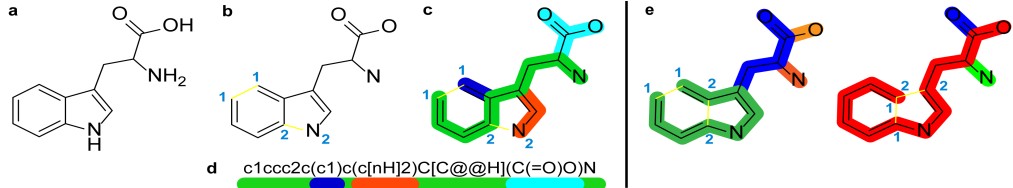

Figure 1: The molecular graph of the amino acid Tryptophan (a). To construct a SMILES string, all cycles are broken, forming a spanning tree (b); a depth-first traversal is selected (c); and this traversal is flattened (d). The beginning and end of intermediate branches in the traversal are denoted by ( and ) respective. The ends of broken cycles are indicated with matching digits. The full grammar is listed in Appendix D. A small set of SMILES strings can cover all paths through a molecule (e).

Molecular property optimization would benefit from a generative model that directly captures the geometry of the space of molecular graphs, rather than SMILES strings, but efficiently infers a latent representation sensitive to spatially distributed molecular features. To this end, we introduce the All SMILES VAE, which uses recurrent neural networks (RNNs) on multiple SMILES strings to implicitly perform efficient message passing along and amongst many flattened spanning trees of the molecular graph in parallel. A fixed-length latent representation is distilled from the variable-length RNN output using attentional mechanisms. From this latent representation, the decoder RNN reconstructs a set of SMILES strings disjoint from those input to the encoder, ensuring that the latent representation only captures features of the molecule, rather than its SMILES realization. Simple property regressors jointly trained on this latent representation surpass the state-of-the-art for molecular property prediction, and facilitate exceptional gradient-based molecular property optimization when constrained to the region of the prior containing almost all the probability around it. We further demonstrate that the latent representation forms a near-bijection with the space of molecules, and is smooth with respect to molecular properties, facilitating effective optimization. For a complete delineation of our novel contributions relative to past work, see Appendix B.4.

## 2 EFFICIENT MOLECULAR ENCODING WITH MULTIPLE SMILES STRINGS

A variational autoencoder (VAE) defines a generative model over an observed space $x$ in terms of a prior distribution over a latent space $p(z)$ and a conditional likelihood of observed states given the latent configuration $p(x|z)$ (Kingma & Welling, 2013; Rezende et al., 2014). The true log-likelihood $\log[p(x)] = \log\left[\int_z p(z)p(x|z)\right]$ is intractable, so the evidence lower bound (ELBO), based upon a variational approximation $q(z|x)$ to the posterior distribution, is maximized instead: $\mathcal{L} = \mathbb{E}_{q(z|x)}[\log p(x|z)] - D_{\mathrm{KL}}[q(z|x)||p(z)]$. The ELBO implicitly defines a stochastic autoencoder, with encoder $q(z|x)$ and decoder $p(x|z)$.

Many effective molecule encoders rely upon graph convolutions: local message passing in the molecular graph, between either adjacent nodes or adjacent edges (Duvenaud et al., 2015; Kearnes et al., 2016; Kipf & Welling, 2016a; Li et al., 2015; Lusci et al., 2013). To maintain permutation symmetry, the signal into each node is a sum of messages from the adjacent nodes, but may be a function of edge type, or attentional mechanisms dependent upon the source and destination nodes (Ryu et al., 2018). This sum of messages is then subject to a linear transformation and a pointwise nonlinearity. Messages are sometimes subject to gating (Li et al., 2015), like in long short-term memories (LSTM) (Hochreiter & Schmidhuber, 1997) and gated recurrent units (GRU) (Cho et al., 2014), as detailed in Appendix B.1.

Message passing on molecular graphs is analogous to a traditional convolutional neural network applied to images (Krizhevsky et al., 2012; LeCun et al., 1990), with constant-resolution hidden layers (He et al., 2016) and two kernels: a $3 \times 3$ average-pooling kernel that sums messages from adjacent pixels (corresponding to adjacent nodes in a molecular graph), and a trainable $1 \times 1$ kernel that transforms the message from each pixel (node) independently, before a pointwise nonlinearity. While such convolutional networks are now standard in the visual domain, they use hundreds of layers to pass information throughout the image and achieve effective receptive fields that span the entire input (Szegedy et al., 2016). In contrast, molecule encoders generally use between three and seven rounds of message passing (Duvenaud et al., 2015; Gilmer et al., 2017; Jin et al., 2018; Kearnes et al., 2016; Liu et al., 2018; Samanta et al., 2018; You et al., 2018). This limits the computational cost, since each iteration of message passing only propagates information a geodesic distance of one within the molecular graph.[1] In the case of the commonly used dataset of 250,000 drug-like molecules (Gómez-Bombarelli et al., 2018), information cannot traverse these graphs effectively, as their average diameter is 11.1, and their maximum diameter is 24, as shown in Appendix A.

Non-local molecular properties, requiring long-range information propagation along the molecular graph, are of practical interest in domains including pharmaceuticals, photovoltaics, and OLEDs. The pharmacological efficacy of a molecule generally depends upon high binding affinity for a particular receptor or other target, and low binding affinity for other possible targets. These binding affinities are determined by the maximum achievable alignment between the molecule's electromagnetic fields and those of the receptor. Changes to the shape or charge distribution in one part of the molecule affect the position and orientation at which it fits best with the receptor, inducing shifts and rotations that alter the binding of other parts of the molecule, and changing the binding affinity (Clayden et al., 2001). Similarly, efficient next-generation OLEDs depend on properties, such as the singlet-triplet energy gap, that are directly proportional to the strength of long-range electronic interactions across the molecule (Im et al., 2017). The latent representation of a VAE can directly capture these non-local, nonlinear properties only if the encoder passes information efficiently across the entire molecular graph.

Analogous to graph convolutions, gated RNNs defined directly on SMILES strings effectively pass messages, via the hidden state, through a flattened spanning tree of the molecular graph (see Figure 1). The message at each symbol in the string is a weighted sum of the previous message and the current input, followed by a pointwise nonlinearity and subject to gating, as reviewed in Appendix B.1. This differs from explicit graph-based message passing in that the molecular graph is flattened into a chain corresponding to a depth-first pre-order traversal of a spanning tree, and the set of adjacent nodes that affect a message only includes the preceding node in this chain. Rather than updating all messages in parallel, RNNs on SMILES strings move sequentially down the chain, so earlier messages influence

---

[1] All-to-all connections allow fast information transfer, but computation is quadratic in graph size (Gilmer et al., 2017; Kearnes et al., 2016). Lusci et al. (2013) considered a set of DAGs rooted at every atom, with full message propagation in a single pass.

all later messages, and information can propagate through all branches of a flattening of a spanning tree in a single pass. With a well-chosen spanning tree, information can pass the entire width of the molecular graph in a single RNN update. The relationship between RNNs on SMILES strings and graph-based architectures is further explored in Appendix B.

# 3 MODEL ARCHITECTURE

To marry the latent space geometry induced by graph convolutions to the information propagation efficiency of RNNs on SMILES strings, the All SMILES encoder combines these architectures. It takes multiple distinct SMILES strings of the same molecule as input, and applies RNNs to them in parallel. This implicitly realizes a representative set of message passing pathways through the molecular graph, corresponding to the depth-first pre-order traversals of the spanning trees underlying the SMILES strings. Between each layer of RNNs, the encoder harmonizes homologous messages between parallel representations of the multiple SMILES strings. In this harmonization, all messages to a single atom across the multiple SMILES strings are replaced with their pooled average, so that information flows along the union of the implicit SMILES pathways.

Initially, the characters of the multiple SMILES strings are linearly embedded, and each string is preprocessed by a BiGRU (Cho et al., 2014), followed by a linear transformation, to produce the layer 0 representation $\mathbf{H}_i^0$ for each SMILES string $i$. For each SMILES string $i$ and layer $l$, $\mathbf{H}_i^l$ is a sequence of vector embeddings, one for each character of the original SMILES string, collectively forming a matrix. The encoder then applies a stack of modules, each of which harmonizes atom representations across SMILES strings, followed by layer norm, concatenation with the linearly embedded SMILES input, and a GRU applied to the parallel representations independently, as shown in Figures 2 and 3.

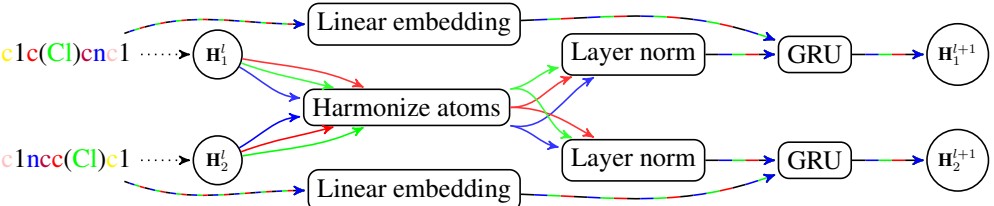

Figure 2: In each layer of the encoder after the initial BiGRU and linear transformation, hidden states corresponding to each atom are harmonized across encodings of different SMILES strings for a common molecule, followed by layer norm and a GRU on each SMILES encoding independently.

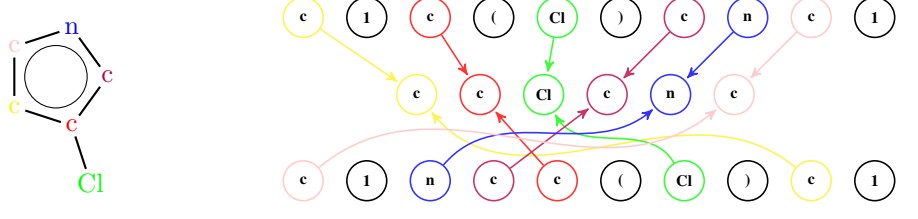

(a) Original molecule      (b) Harmonization of two SMILES strings representing the same molecule

Figure 3: To pass information between multiple SMILES representations of a molecule, the encoder harmonizes the representation of each atom. Homologous messages corresponding to the same atom are pooled (b), and the original messages are replaced with this pooled message, reversing the information flow of (b).

Multiple SMILES strings representing a single molecule need not have the same length, and syntactic characters indicating branching and ring closures rather than atoms and bonds do not generally match. However, the set of atoms is always consistent, and a bijection can be defined between homologous atom characters. At the beginning of each encoder module (Figure 2), the parallel

inputs corresponding to a single, common atom of the original molecule are pooled, as shown in Figure 3. This harmonized atom representation replaces the original in each of the input streams for the subsequent layer normalizations and GRUs, reversing the information flow of Figure 3. To realize atom harmonization, we experimented with average and max pooling, but found element-wise sigmoid gating to be most effective (Dauphin et al., 2017; Li et al., 2015; Ryu et al., 2018): $a' = \frac{1}{k}\sum_k \left(a_k \odot \sigma\left(W\left[a_k, \frac{1}{k}\sum_k a_k\right] + b\right)\right)$, where $[x, y]$ is the concatenation of vectors $x$ and $y$, and the logistic function $\sigma(x)$ is applied element-wise. The pooling effectively sums messages propagated from many adjacent nodes in the molecular graph, analogous to a graph convolution, but the GRUs efficiently transfer information through many edges in each layer, rather than just one. The hidden representations associated with non-atom, syntactic input characters, such as parentheses and digits, are left unchanged by the harmonization operation.

The approximating posterior distills the resulting variable-length encodings into a fixed-length hierarchy of autoregressive Gaussian distributions (Rolfe, 2016). The mean and log-variance of the first layer of the approximating posterior, $\mathbf{z}_1$, is parametrized by max-pooling the terminal hidden states of the final encoder GRUs, followed by batch renormalization (Ioffe, 2017) and a linear transformation, as shown in Figure 4. Succeeding hierarchical layers use Bahdanau-style attention (Bahdanau et al., 2014, reviewed in Appendix B.2) over the pooled final atom vectors, with the query vector defined by a one-hidden-layer network of rectified linear units (ReLUs) given the concatenation of the previous latent layers as input. This is analogous to the order-invariant encoding of set2set, but an output is produced at each step, and processing is not gated (Vinyals et al., 2015). The attentional mechanism is also effectively available to property regressors that take the fixed-length latent representation as input, allowing them to aggregate contributions from across the molecule. The output of the attentional mechanism is subject to batch renormalization and a linear transformation to compute the conditional mean and log-variance of the layer. The prior has a similar autoregressive structure, but uses neural networks of ReLUs in place of Bahdanau-style attention, as there are no atom vectors. This is illustrated in Appendix B in Figure 11, briefly, each layer of the hierarchy is a two layer neural network conditioned on the previous latent samples, that is, $p_\theta(z_i, .., z_n) = p(z_1) \prod_{i=2}^n p_{\theta_i}(z_i|z_{<i})$. For molecular optimization tasks, we usually scale up the KL-term in the ELBO by the number of SMILES strings in the decoder, analogous to multiple single-SMILES VAEs in parallel; we leave this KL term unscaled for property prediction.

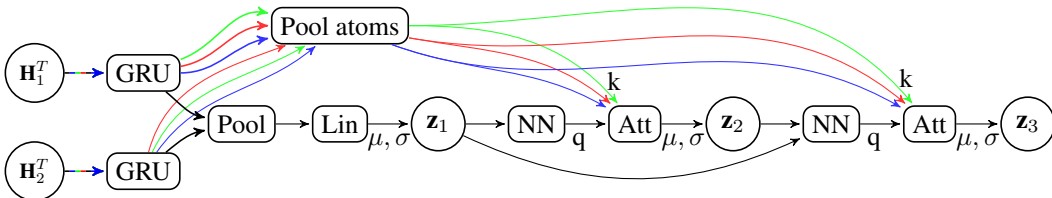

Figure 4: The approximating posterior is an autoregressive set of Gaussian distributions. The mean ($\mu$) and log-variance ($\log \sigma^2$) of the first subset of latent variables $\mathbf{z}_1$ is a linear transformation of the max-pooled final hidden state of GRUs fed the encoder outputs. Succeeding subsets $\mathbf{z}_i$ are produced via Bahdanau-style attention with the pooled atom outputs of the GRUs as keys ($k$), and the query ($q$) computed by a neural network on $\mathbf{z}_{<i}$.

The decoder is a single-layer LSTM, for which the initial cell state is computed from the latent representation $\mathbf{z} = [\mathbf{z}_1, \mathbf{z}_2, \cdots]$ by a neural network, and a linear transformation of the latent representation is concatenated onto each input. It is trained with teacher forcing to reconstruct a set of SMILES strings disjoint from those provided to the encoder, but representing the same molecule. As in conventional language models, the decoder LSTM autoregressively produces a sequence of categorical distributions for each successive SMILES character conditioned on the preceding characters. We decode multiple SMILES in parallel with shared decoder parameters, conditioned on the same latent representation. When decoding any SMILES from the same molecule, the decoder may predict characters with low confidence, but as the decoding proceeds, the decoder is conditioned on previous characters of the SMILES string, becoming more certain about which string is meant to be decoded. Grammatical constraints (Dai et al., 2018; Kusner et al., 2017) can naturally be enforced by parsing the unfolding character sequence with a pushdown automaton, and constraining the final

softmax of the LSTM output at each time step to grammatically valid symbols. This is detailed in Appendix D, although we leave the exploration of this technique to future work.

The All SMILES VAE is a generative model over both the structure and properties of molecules $\mathcal{M}$, so we define the conditional likelihood to be $p(\mathcal{M}|z) = p\left(\rho^{\mathcal{M}}|z\right) \cdot \prod_j p\left(x_j^{\mathcal{M}}|z\right)$, where $\left\{x_j^{\mathcal{M}}\right\}_{j=1}^N$ is a set of $N$ SMILES strings of a molecule $\mathcal{M}$ with properties $\rho^{\mathcal{M}}$. Unlike a conventional VAE, the representation of the molecule $\mathcal{M}$ input to the encoder $q(z|\mathcal{M})$ is not identical to the target of the conditional likelihood $p(\mathcal{M}|z)$; rather, it comprises a set of SMILES strings $\left\{x_i^{\mathcal{M}}\right\}_{i=1}^M$ of the molecule $\mathcal{M}$ disjoint from the decoding target, and does not include the molecular properties. Nevertheless, both encoder input and decoder target correspond to a single molecule $\mathcal{M}$. The conditional log likelihood of the molecular properties $\log p\left(\rho^{\mathcal{M}}|z\right)$ is implicitly parametrized by scaling its contribution to the ELBO by $\lambda$. For instance, if $p\left(\rho^{\mathcal{M}}|z\right)$ is a unit-variance Gaussian distribution, then $\lambda$ sets the effective variance to $\lambda^{-1}$. Finally, when optimizing molecular properties, we scale the KL term by $M$, the number of SMILES strings in the decoder, rendering the ELBO analogous to multiple single-SMILES VAEs in parallel. The resulting ELBO is:

$$\mathcal{L} = \mathbb{E}_{z \sim q\left(z|\{x_i\}_{i=1}^N\right)} \left[ \lambda \cdot \log p(\rho|z) + \sum_{j=1}^M \log p(x_j|z) \right] + M \cdot D_{\mathrm{KL}}\left[ q\left(z|\{x_i\}_{i=1}^N\right) \| p_\theta(z) \right].$$

In our experiments we parametrize the regressors and classifiers used in $p(\rho|z)$ by a single linear layer, with an activation function to put the predicted property into its correct range. Since the SMILES inputs to the encoder are different from the targets of the decoder, the decoder is effectively trained to assign high probability to all SMILES strings of the encoded molecule. The latent representation must capture the molecule as a whole, rather than any particular SMILES input to the encoder. To accommodate this intentionally difficult reconstruction task, facilitate the construction of a bijection between latent space and molecules, and following prior work (Kang & Cho, 2018; Winter et al., 2019a), we use a width-5 beam search decoder to map from the latent representation to the space of molecules at test-time. In all experiments, we use a set of $M = 5$ randomly selected SMILES strings for encoding, and $N = 5$ disjoint SMILES strings as the decoding target. Further architectural details are presented in Appendix B.

## 3.1 COMPUTATIONAL COMPLEXITY

Since the length of a SMILES string is linear in the total number of bonds $b$, the computational complexity of each layer of the All SMILES encoder is $\mathcal{O}(M \cdot b)$, where $M = 5$ is the number of random SMILES strings of the molecule. Similarly, the complexity of each layer of graph convolution is $\mathcal{O}(b)$. However, to pass information through the entire molecule, graph convolutions require a number of layers proportional to the graph diameter. Molecular graph convolutions generally use a fixed architecture for all molecules. In principle, the maximum diameter of a molecule is equal to the number of bonds. As a result, the computational complexity for graph convolutions to pass information through all molecules is $\mathcal{O}\left(b^2\right)$. In contrast, each RNN in the All SMILES encoder can in principle pass information through the entire graph, so the computational complexity remains $\mathcal{O}(M \cdot b)$. This analysis however does not take into account the practical speed up of sparse parallelizable graph convolutional operations in comparison with the sequential nature of RNNs.

## 3.2 LATENT SPACE OPTIMIZATION

Unlike many models that apply a sparse Gaussian process to fixed latent representations to predict molecular properties (Dai et al., 2018; Jin et al., 2018; Kusner et al., 2017; Samanta et al., 2018), the All SMILES VAE jointly trains property regressors with the generative model (as do Liu et al., 2018).[2] We use linear regressors for the log octanol-water partition coefficient (logP) and molecular weight (MW), which have unbounded values; and logistic regressors for the quantitative estimate of drug-likeness (QED) (Bickerton et al., 2012) and twelve binary measures of toxicity (Huang et al., 2016; Mayr et al., 2016), which take values in $[0, 1]$. We then perform gradient-based optimization of the property of interest with respect to the latent space, and decode the result to produce an optimized molecule.

---

[2]Gómez-Bombarelli et al. (2018) jointly train a regressor, but still optimize using a Gaussian process.

Naively, we might either optimize the predicted property without constraints on the latent space, or find the maximum a posteriori (MAP) latent point for a conditional likelihood over the property that assigns greater probability to more desirable values. However, the property regressors and decoder are only accurate within the domain in which they have been trained: the region assigned high probability mass by the prior. For a $n$-dimensional standard Gaussian prior, almost all probability mass lies in a practical support comprising a thin spherical shell of radius $\sqrt{n-1}$ (Blum et al., 2017, Gaussian Annulus Theorem). With linear or logistic regressors, predicted property values increase monotonically in the direction of the weight vector, so unconstrained property maximization diverges from the origin of the latent space. Conversely, MAP optimization with a Gaussian prior is pulled towards the origin, where the density of the prior is greatest. Both unconstrained and MAP optimization thus deviate from the practical support in each layer of the hierarchical prior, resulting in large prediction errors and poor optimization.

We can use the reparametrization trick (Kingma & Welling, 2013; Rezende et al., 2014) to map our autoregressive prior back to a standard Gaussian. The image of the thin spherical shell through this reparametrization still contains almost all of the probability mass. We therefore constrain optimization to the reparametrized $n-1$ dimensional sphere of radius $\sqrt{n-1}$ for each $n$-dimensional layer of the hierarchical prior by optimizing the angle directly.[3] Although the reparametrization from the standard Gaussian prior to our autoregressive prior is not volume preserving, this hierarchical radius constraint holds us to the center of the image of the thin spherical shell. The distance to which the image of the thin spherical shell extends away from the $n-1$ dimensional sphere at its center is a highly nonlinear function of the previous layers. We describe this hierarchical radius constraint in more detail, and provide pseudocode, in Appendix B.3.

## 4 RESULTS

We evaluate the All SMILES VAE on standard 250,000 and 310,000 element subsets (Gómez-Bombarelli et al., 2018; Kang & Cho, 2018) of the ZINC database of small organic molecules (Irwin et al., 2012; Sterling & Irwin, 2015). We also evaluate on the Tox21 dataset (Huang et al., 2016; Mayr et al., 2016) in the DeepChem package (Wu et al., 2018), comprising binarized binding affinities of 7831 compounds against 12 proteins. For further details, see Appendix A. Additional experiments, including ablations of novel model components, are described in Appendix C.

The full power of continuous, gradient-based optimization can be brought to bear on molecular properties given a bijection between molecules and contractible regions of a latent space, along with a regressor from the latent space to the property of interest that is differentiable almost everywhere. Such a bijection is challenging to confirm, since it is difficult to find the full latent space preimage of a molecule implicitly defined by a mapping from latent space to SMILES strings, such as our beam search decoder. As a necessary condition, we confirm that it is possible to map from the space of molecules to latent space and back again, and that random samples from the prior distribution in the latent space map to valid molecules. The former is required for injectivity, and the latter for surjectivity, of the mapping from molecules to contractible regions of the latent space.

Using the approximating posterior as the encoder, but always selecting the mean of each conditional Gaussian distribution, and using a beam search over the conditional likelihood as the decoder, $87.4\% \pm 1\%$ of a held-out test set of ZINC250k (80/10/10 train/val/test split) is reconstructed accurately. With the same beam search decoder, $98.5\% \pm 0.1\%$ of samples from the prior decode to valid SMILES strings. We expect that enforcing grammatical constraints in the decoder LSTM, as described in Appendix D, would further increase these rates. All molecules decoded from a set of 50,000 independent samples from the prior were unique, $99.958\%$ were novel relative to the training dataset, and their average synthetic accessibility score (Ertl & Schuffenhauer, 2009) was $2.97 \pm 0.01$, compared to $3.05$ in the ZINC250k dataset used for training.

### 4.1 PROPERTY PREDICTION

Ultimately, we would like to optimize molecules for complicated physical properties, such as fluorescence quantum yield, binding affinity to selected receptors, and low toxicity. Networks can only be trained to predict such physical properties if their true values are known on an appropriate

---

[3]This generalizes the slerp interpolations of Gómez-Bombarelli et al. (2018) to optimization.

training dataset. While simple properties can be accurately computed from first principles, properties like drug efficacy arise from highly nonlinear, poorly characterized processes, and can only be accurately determined through time-consuming and expensive experimental measurements. Since such experiments can only be performed on a small number of molecules, we evaluate the ability of the All SMILES VAE to perform semi-supervised property prediction.

As Figure 5 and Table 5 in Appendix C demonstrate, we significantly improve the state-of-the-art in the semi-supervised prediction of simple molecular properties, including the log octanol-water partition coefficient (logP), molecular weight (MW), and quantitative estimate of drug-likeness (QED) (Bickerton et al., 2012), against which many algorithms have been benchmarked. We achieve a similar improvement in fully supervised property prediction, as shown in Table 1, where we compare to extended connectivity fingerprints (ECFP; Rogers & Hahn, 2010), the character VAE (CVAE; Gómez-Bombarelli et al., 2018), and graph convolutions (Duvenaud et al., 2015). Table 4 in Appendix C documents an even larger improvement compared to models that use a sparse Gaussian process for property prediction. We also surpass the state-of-the-art in toxicity prediction on the Tox21 dataset (Huang et al., 2016; Mayr et al., 2016), as shown in Table 1, despite refraining from ensembling our model, or engineering features using expert chemistry knowledge, as in previous state-of-the-art methods (Zaslavskiy et al., 2019).

Accurate property prediction only facilitates effective optimization if the true property value is smooth with respect to the latent space. In Figure 6a, we plot the true (not predicted) logP over a densely sampled 2D slice of the latent space, where the y axis is aligned with the logP linear regressor.

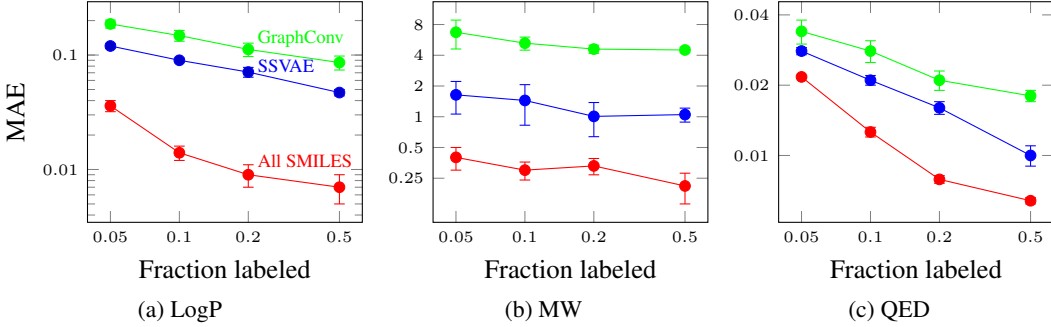

|   (a) LogP   |   (b) MW   |   (c) QED   |

Figure 5: Semi-supervised mean absolute error (MAE) $\pm$ the standard deviation across ten replicates for the log octanol-water partition coefficient (a), molecular weight (b), and the quantitative estimate drug-likeness (c; Bickerton et al., 2012) on the ZINC310k dataset. Plots are log-log; the All SMILES MAE is a fraction of that of the SSVAE (Kang & Cho, 2018) and graph convolutions (Kearnes et al., 2016). Semi-supervised VAE (SSVAE) and graph convolution results are those reported by Kang & Cho (2018).

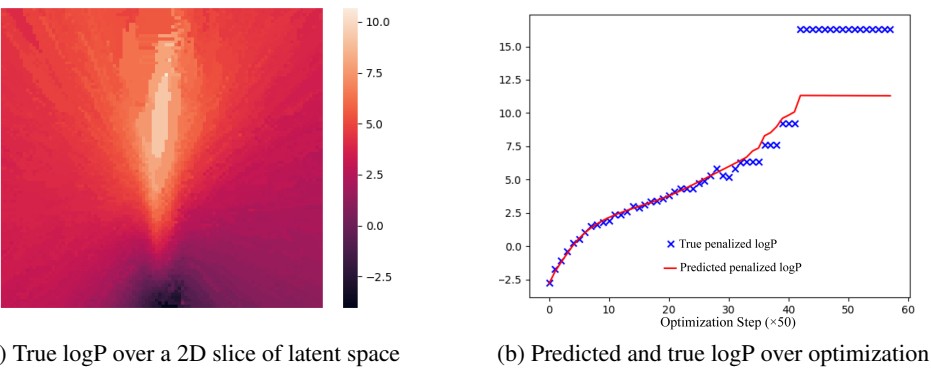

(a) True logP over a 2D slice of latent space     (b) Predicted and true logP over optimization

Figure 6: Dense decodings of true logP along a local 2D sheet in latent space, with the y axis aligned with the regressor (a), and predicted and true penalized logP across steps of optimization (b).

Table 1: Fully supervised regression on ZINC250k (a), evaluated using the mean absolute error; and Tox21 (b), evaluated with the area under the receiver operating characteristic curve (AUC-ROC), averaged over all 12 toxicity types. Aside from All SMILES, results are those reported by ECFP: (Rogers & Hahn, 2010), CVAE: (Gómez-Bombarelli et al., 2018), GraphConv: (Duvenaud et al., 2015), Graph Conv + Super Node (SN): (Li et al., 2017), PotentialNet: (Feinberg et al., 2018), and ToxicBlend: (Zaslavskiy et al., 2019). The ablation of atom harmonization is also evaluated on the Tox21 dataset.

| (a) ZINC250k | | | | (b) Tox21 | |
|---|---|---|---|---|---|
| MODEL | MAE LOGP | MAE QED | | MODEL | AUC-ROC |
| ECFP | 0.38 | 0.045 | | GRAPHCONV + SN | 0.854 |
| CVAE | 0.15 | 0.054 | | POTENTIALNET | $0.857_{\pm 0.006}$ |
| CVAE ENC | 0.13 | 0.037 | | TOXICBLEND | 0.862 |
| GRAPHCONV | 0.05 | 0.017 | | **All SMILES (no harmonization)** | $\mathbf{0.864}_{\pm 0.003}$ |
| **All SMILES** | $\mathbf{0.005}_{\pm 0.0006}$ | $\mathbf{0.0052}_{\pm 0.0001}$ | | **All SMILES** | $\mathbf{0.8751}_{\pm 0.0008}$ |

## 4.2 MOLECULAR OPTIMIZATION

We maximize the output of our linear and logistic property regressors, plus a log-prior regularizer, with respect to the latent space, subject to the hierarchical radius constraint described in Section 3.2. After optimizing in the latent space with ADAM, we project back to a SMILES representation of a molecule with the decoder. Following prior work, we optimize QED and logP penalized by the synthetic accessibility score and the number of large rings (Dai et al., 2018; Jin et al., 2018; Kusner et al., 2017; Samanta et al., 2018; You et al., 2018; Zhou et al., 2018). Figure 6b depicts the predicted and true logP over an optimization trajectory, while Table 2 compares the top three values found amongst 100 such trajectories to the previous state-of-the-art.[4] The molecules realizing these property values are shown in Figure 7. Leaving the KL term in the ELBO unscaled by the number of SMILES strings in the decoder reduces the regularization of the latent space embeddings, allowing latent space optimization to search a wider space of molecules that are less similar to the training set, as shown in Figure 14 of Appendix C.3.

Table 2: Properties of the top three optimized molecules trained on ZINC250k. Other results are taken from JT-VAE: (Jin et al., 2018), GCPN: (You et al., 2018), MolDQN: (Zhou et al., 2018), and CGVAE: (Liu et al., 2018). Following prior work, penalized logP is normalized by the statistics of the Zinc250k dataset.

| MODEL | PENALIZED LOGP | | MODEL | QED |
|---|---|---|---|---|
| JT-VAE | 5.30, 4.93, 4.49 | | JT-VAE | 0.925, 0.911, 0.910 |
| GCPN | 7.98, 7.85, 7.80 | | CGVAE | 0.938, 0.931, 0.880 |
| MOLDQN | 8.93, 8.93, 8.91 | | GCPN | 0.948, 0.947, 0.946 |
| ALL SMILES | 12.31, 12.13, 12.01 | | **MolDQN** | **0.948, 0.948, 0.948** |
| **All SMILES (KL unscaled)** | **29.80, 29.76, 29.11** | | **All SMILES** | **0.948, 0.948, 0.948** |

## 4.3 ABLATION OF MODEL COMPONENTS

In Table 3, we progressively ablate model components to demonstrate that all elements of the All SMILES architecture contribute to building a powerful fixed-length representation of molecules, rather than their particular SMILES string instantiations. We evaluate the effect of these ablations on the mean absolute error (MAE) of logP and QED predictions, as well as the percentage of samples

---

[4]Zhou et al. (2018) appear to report unnormalized penalized logP values: 11.84, 11.84, 11.82. In Table 2, we recompute normalized values for their best molecules. Recently, Winter et al. (2019b) reported molecules with penalized logP as large as 26.1, but train on an enormous, non-standard dataset of 72 million compounds aggregated from the ZINC15 and PubChem databases.

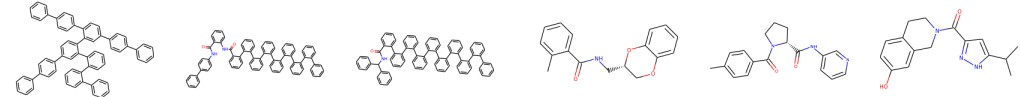

(a) Molecules with the top three penalized logP values

(b) Molecules with the top three QED values

Figure 7: Molecules produced by gradient-based optimization in the All SMILES VAE.

from the prior that decode to valid SMILES strings (Val) and the percentage of test molecules that are reconstructed accurately (Rec acc). In all cases, we use the the mean of each conditional Gaussian distribution, and a beam-search decoder.

NO ATOM HARMONIZATION removes the pooling amongst each instance of an atom across SMILES strings in the encoder, depicted in Figure 3. As a result, the multiple SMILES inputs are processed independently until the final max pooling over GRU hidden states. A random SMILES string is chosen to serve as input to the attention mechanisms of the approximating posterior. Table 1b shows the significant effect of this ablation on toxicity prediction, demonstrating the importance of atom harmonization for nonlinear properties of the entire molecule, in contrast to the quasi-linear logP and QED reported in Table 3. We extend this process in ONE SMILES ENC by only feeding a single SMILES string to the encoder, although the decoder still reconstructs multiple disjoint SMILES strings. ONE SMILES ENC/DEC ($\neq$) further reduces the size of the decoder set to one, but the encoded and decoded SMILES strings are distinct. Finally, ONE SMILES ENC/DEC ($=$) encodes and decodes a single, shared SMILES string. Except for ONE SMILES ENC/DEC ($=$), all of these ablations primarily disrupt the flow of messages between the flattened spanning trees, and induce a similar, significant decay in performance. ONE SMILES ENC/DEC ($=$) further permits the latent representation to encode the details of the particular SMILES string, rather than forcing the representation of only the underlying molecule, and causes a further reduction in performance. This ablation uses canonical SMILES for encoding and decoding, as in Gómez-Bombarelli et al. (2018).

We also observe a meaningful contribution from the hierarchical approximating posterior. In NO POSTERIOR HIERARCHY, we move all latent variables to the first layer of the hierarchy, removing the succeeding layers. The prior is a standard Gaussian, and there is no attentional pooling over the atom representations. Further ablations of the hierarchical radius constraint are reported in Appendix C.4.

Table 3: Effect of model ablation on fully supervised property prediction and generative modeling using the ZINC250k dataset.

| ABLATION | MAE LOGP | MAE QED | VAL | REC ACC |
|---|---|---|---|---|
| FULL MODEL | 0.005±0.0006 | 0.0052±0.0001 | 98.5±0.1 | 87.4±1.0 |
| NO ATOM HARMONIZATION | 0.008±0.004 | 0.0076±0.0005 | 97.6±0.2 | 84.0±0.4 |
| ONE SMILES ENC | 0.008±0.005 | 0.0073±0.0002 | 98.4±0.1 | 82.3± 0.4 |
| ONE SMILES ENC/DEC ($\neq$) | 0.009±0.001 | 0.0091±0.0003 | 97.1±0.7 | 80.9±0.4 |
| ONE SMILES ENC/DEC ($=$) | 0.025±0.003 | 0.0115±0.0004 | 85.7±1 | 91.3±0.6 |
| NO POSTERIOR HIERARCHY | 0.010±0.003 | 0.0051±0.0001 | 98.2±0.5 | 85.2±0.6 |

## 5 CONCLUSION

For each molecule, the All SMILES encoder uses stacked, pooled RNNs on multiple SMILES strings to efficiently pass information throughout the molecular graph. The decoder targets a disjoint set of SMILES strings of the same molecule, forcing the latent space to develop a consistent representation for each molecule. Attentional mechanisms in the approximating posterior summarize spatially diffuse features into a fixed-length, non-factorial approximating posterior, and construct a latent representation on which linear regressors achieve state-of-the-art semi- and fully-supervised property prediction. Gradient-based optimization of these regressor outputs with respect to the latent representation, constrained to a subspace near almost all probability in the prior, produces state-of-the-art optimized molecules when coupled with a simple RNN decoder.

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

# A DATASETS

SMILES strings, as well as the true values of the log octanol-water partition coefficient (logP), molecular weight (MW), and the quantitative estimate of drug-likeness (QED), are computed using RDKit (Landrum et al., 2006).

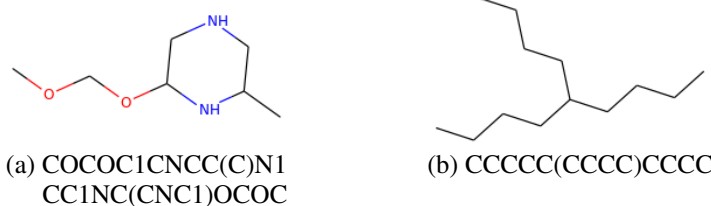

(a) COCOC1CNCC(C)N1
CC1NC(CNC1)OCOC

(b) CCCCC(CCCC)CCCC

Figure 8: Multiple SMILES strings of a single molecule may be more dissimilar than SMILES strings of radically dissimilar molecules. The top SMILES string for molecule (a) is 30% similar to the bottom SMILES string by string edit distance, but 60% similar to the SMILES string for molecule (b).

## A.1 ZINC

For molecular property optimization and fully supervised property prediction, we train the All SMILES VAE on the ZINC250k dataset of 250,000 organic molecules with between 6 and 38 heavy atoms, and penalized logPs[5] from -13 to 5 (Gómez-Bombarelli et al., 2018). This dataset is curated from a subset of the ZINC12 dataset (Irwin et al., 2012), and available from `https://github.com/aspuru-guzik-group/chemical_vae`. The distribution of molecular diameters in ZINC250k is shown in Figure 9.

For semi-supervised property prediction on logP, MW, and QED, we train on the ZINC310k dataset of 310,000 organic molecules with between 6 and 38 heavy atoms (Kang & Cho, 2018). This dataset is curated from the full ZINC15 dataset (Sterling & Irwin, 2015), and available from `https://github.com/nyu-dl/conditional-molecular-design-ssvae`.

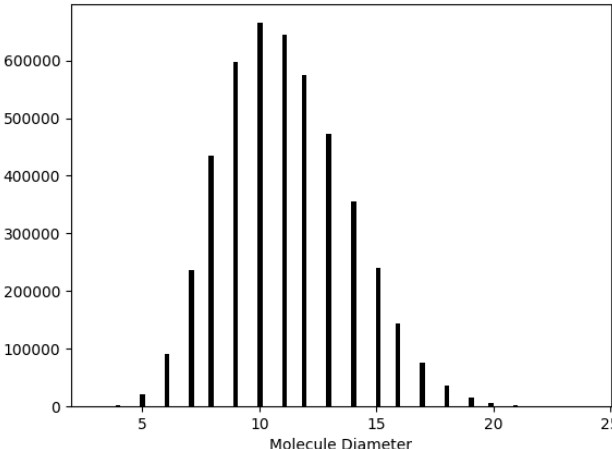

Figure 9: Histogram of molecular diameters in the ZINC250k dataset. The diameter is defined as the maximum eccentricity over all atoms in the molecular graph. The mean is 11.1; the maximum is 24. Typical implementations of graph convolution use only three to seven rounds of message passing (Duvenaud et al., 2015; Gilmer et al., 2017; Jin et al., 2018; Kearnes et al., 2016; Liu et al., 2018; Samanta et al., 2018; You et al., 2018), and so cannot propagate information across most molecules in this dataset.

---

[5]The log octanol-water partition coefficient minus the synthetic accessibility score and the number of rings with more than six atoms.

## A.2 Tox21

For the semi-supervised prediction of twelve forms of toxicity, we train on the Tox21 dataset (Huang et al., 2016; Mayr et al., 2016), accessed through the DeepChem package (Wu et al., 2018), with the provided random train/validation/test set split. This dataset contains binarized binding affinities against up to 12 proteins for 6264 training, 783 validation, and 784 test molecules. Tox21 contains molecules with up to 140 atoms, ranging from large peptides to lanthanide, actinide and other metals. Many of these metal atoms are not present in any of the standard molecular generative modeling datasets, and there are metal atoms in the validation and test set that never appear in the training set. To address these difficulties, we curated an unsupervised dataset of 1.5 million molecules from the PubChem database (Kim et al., 2018), which we will make available upon publication. To maintain commensurability with prior work, this additional unsupervised dataset is *only* used on the Tox21 prediction task.

## B    Extended Model Architecture

The full All SMILES VAE architecture is summarized in Figure 10. The evidence lower bound (ELBO) of the log-likelihood,

$$\mathcal{L} = \mathbb{E}_{q(z|x)} \left[\log p(x|z)\right] - D_{\mathrm{KL}} \left[q(z|x)||p(z)\right], \tag{1}$$

is the sum of the conditional log-likelihoods of $\mathbf{x}'_i$ in Figure 10, minus the Kullback-Leibler divergence between the approximating posterior, $q(z|x)$, computed by node AP in Figure 10, and the prior depicted in Figure 11.

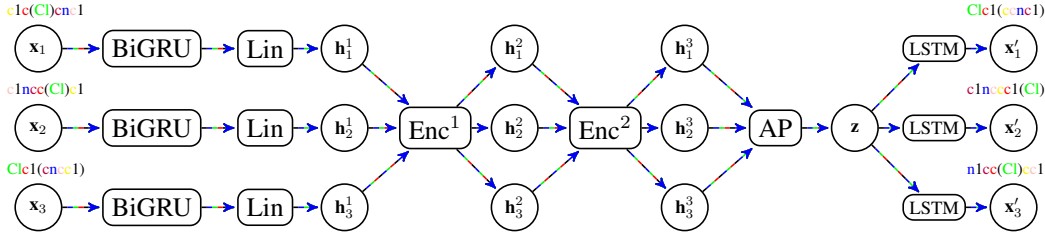

Figure 10: Multiple SMILES strings representing a single, common molecule are preprocessed by a BiGRU and a linear transformation, followed by multiple encoder blocks as in Figures 2 and 3. The approximating posterior depicted in Figure 4 then produces a sample from the latent state $\mathbf{z}$, which is decoded into SMILES strings by LSTMs. Note that all SMILES strings, in both the input and the output, are distinct. The encoder blocks also receive a linear embedding of the original SMILES strings as input.

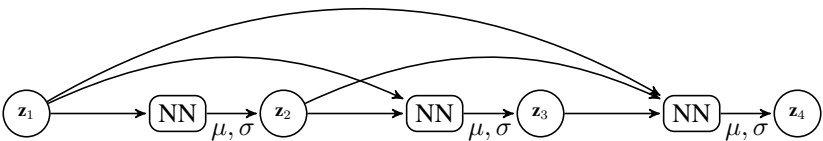

Figure 11: The prior distribution over $\mathbf{z} = [\mathbf{z_1}, \mathbf{z_2}, \cdots]$ is a hierarchy of autoregressive Gaussians. The conditional prior distribution of hierarchical layer $i$ given layers 1 through $i-1$, $p(\mathbf{z}_i|\mathbf{z}_1, \mathbf{z}_2, \cdots \mathbf{z}_{i-1})$, is a Gaussian with mean $\mu$ and log-variance $\log \sigma^2$ determined by a neural network with input $[\mathbf{z}_1, \mathbf{z}_2, \cdots, \mathbf{z}_{i-1}]$.

In all experiments, we use encoder stacks of depth three, with 512 hidden units in each GRU. The approximating posterior uses four layers of hierarchy, with 128 hidden units in the one-hidden-layer neural network that computes the attentional query vector. In practice, separate GRUs were used to produce the final hidden state for $\mathbf{z}_1$ and the atom representations for $\mathbf{z}_{>1}$. The single-layer LSTM decoder has 2048 hidden units. Training was performed using ADAM, with a decaying learning

rate and KL annealing. In all multiple SMILES strings architectures, we use 5 SMILES strings for encoding and decoding which are selected with RDkit (Landrum et al., 2006).

In contrast to many previous molecular VAEs, we do not scale down the term $D_{\mathrm{KL}}\left[q(z|x)||p(z)\right]$ in the ELBO by the number of latent units (Dai et al., 2018; Kusner et al., 2017). However, our loss function does include separate reconstructions for multiple SMILES strings of a single molecule. For molecular optimization tasks, we usually scale up this KL term by the number of SMILES strings in the decoder, analogous to multiple single-SMILES VAEs in parallel; we leave the KL term unscaled for property prediction.

### B.1 GATED RECURRENT NEURAL NETWORKS

Convolutional neural networks on images (Krizhevsky et al., 2012; LeCun et al., 1990) leverage the inherent geometry of the visual domain to perform local message passing. At every spatial location of each layer, a convolutional network computes a message comprising the weighted sum of messages from the surrounding region in the preceding layer, followed by a point-wise nonlinearity. Each round of messages propagates information a distance equal to the convolutional kernel diameter multiplied by the current spatial resolution.

Recurrent neural networks, such as long short-term memories (LSTMs) (Hochreiter & Schmid-huber, 1997) and gated recurrent units (GRUs) (Cho et al., 2014), model text, audio, and other one-dimensional sequences in an analogous manner. The kernel, comprising the weights on the previous hidden state and the current input, has a width of only two. However, the messages (i.e., the hidden states) are updated consecutively along the sequence, so information can propagate through the entire network in a single pass, substantially reducing the number of layers required. LSTMs and GRUs are ubiquitous in natural language processing tasks, and efficient GPU implementations have been developed (Appleyard et al., 2016).

Gated recurrent units (GRUs) are defined by (Cho et al., 2014):

$$[r, z] = \sigma\left(x_t\left[W_r, W_z\right] + h_{t-1}\left[U_r, U_z\right] + \left[b_r, b_z\right]\right)$$
$$h_t = (1 - z) \odot h_{t-1} + z \odot \tanh\left(x_t W + (r \odot h_{t-1}) U + b_h\right)$$

where $r$, $z$, and $h$ are row-vectors, $[x, y]$ denotes the column-wise concatenation of $x$ and $y$, and the logistic function $\sigma(x) = \left(1 + e^{-x}\right)^{-1}$ and hyperbolic tangent are applied element-wise to vector argument $x$. The hidden state $h_t$, comprising the message from node $t$, is a gated, weighted sum of the previous message $h_{t-1}$ and the current input $x_t$, both subject to an element-wise linear transformation and nonlinear (sigmoid) transformation. Specifically, the sum of the message from the input, $x_t W U^{-1}$, and the gated message from the previous node, $r \odot h_{t-1}$, is subject to a linear transformation $U$ and a pointwise nonlinearity. This is then gated and added to a gated residual connection from the previous node.

Long short-term memories (LSTMs) are defined similarly (Hochreiter & Schmidhuber, 1997):

$$[f_t, i_t, o_t] = \sigma\left(x_t[W_f, W_i, W_o] + h_{t-1}[U_f, U_i, U_o] + [b_f, b_i, b_o]\right)$$
$$c_t = f_t \odot c_{t-1} + i_t \odot \tanh\left(x_t W_c + h_{t-1} U_c + b_c\right)$$
$$h_t = o_t \odot \tanh\left(c_t\right)$$

where $f$ is the forget gate, $i$ is the input gate, and $o$ is the output gate. LSTMs impose a second hyperbolic tangent and gating unit on the nonlinear recurrent message, but nevertheless still follow the form of applying width-two kernels and pointwise nonlinearities to the input and hidden state.

In contrast, message passing in graphs is defined by (Duvenaud et al., 2015; Kearnes et al., 2016; Kipf & Welling, 2016a; Li et al., 2015):

$$h_t^{(n)} = f\left(\left(\sum_{m \in \mathcal{N}(n)} h_{t-1}^{(m)}\right) W_t\right)$$

where $\mathcal{N}(n)$ is the set of neighbors of node $n$, for which there is an edge between $n$ and $m \in \mathcal{N}(n)$, and $f(x)$ is a pointwise nonlinearity such as a logistic function or rectified linear unit. This message passing, also called graph convolutions, can be understood as a first-order approximation to spectral

convolutions on graphs (Hammond et al., 2011). (Kipf & Welling, 2016a) additionally normalize each message by the square root of the degree of each node before and after the sum over neighboring nodes. (Kearnes et al., 2016) maintain separate messages for nodes and edges, with the neighborhood of a node comprising the connected edges and the neighborhood of an edge comprising the connected nodes. (Li et al., 2015) add gating analogous to a GRU.

An LSTM, taking a SMILES string as input, can realize a subset of the messages passed by graph convolutions. For instance, input gates and forget gates can conspire to ignore open-parentheses, which indicate the beginning of a branch of the depth-first spanning tree traversal. If they similarly ignore the digits that close broken rings, the messages along each branch of the flattened spanning tree are not affected by the extraneous SMILES syntax. Input and forget gates can then reset the LSTM's memory at close-parentheses, which indicate the end of a branch of the depth-first spanning tree traversal, and the return to a previous node, ensuring that messages only propagate along connected paths in the molecular graph.

A set of LSTMs on multiple SMILES strings of a single molecule, with messages exchanged between the LSTMs, can generate all of the messages produced by a graph convolution. Atom-based pooling between LSTMs on multiple SMILES strings of the same molecule combines the messages produced in each flattened spanning tree, allowing every LSTM to access all messages produced by a graph convolution. While an LSTM decoder generating SMILES strings faces ambiguity regarding which set of SMILES strings representing a molecule to produce, this is analogous to the problem faced by graph-based decoders, as discussed in Appendix D.2

### B.2    BAHDANAU-STYLE ATTENTION

The layers of the hierarchical approximating posterior after the first define a conditional Gaussian distribution, the mean and log-variance of which are parametrized by an attentional mechanism of the form proposed by (Bahdanau et al., 2014). The final encoder hidden vectors for each atom comprise the key vectors $k$, whereas the query vector $q$ is computed by a one-hidden-layer network of rectified linear units given the concatenation of the previous latent layers as input. The final output of the attentional mechanism, $c$, is computed via:

$$e_i = \tanh\left(qW_a + k_iU_a\right)v^\top$$
$$\alpha_i = \frac{\exp(e_i)}{\sum_j \exp(e_j)}$$
$$c = \sum_i \alpha_i k_i$$

The output of the attentional mechanism is subject to batch renormalization and a linear transformation to compute the conditional mean and log-variance of the layer.

### B.3    LATENT SPACE OPTIMIZATION

As discussed in the text in section 3.2, optimization in the latent space is performed in a region of space constrained to a sphere, chosen so that the space around it contains almost all of the probability mass of the prior. This radius of the sphere is defined by the prior distribution, and is equal to $\sqrt{N-1}$ for a standard Gaussian. However, due to our nonlinear, autoregressive parametrization of the prior, this radius is not analytically accessible, and so we cannot apply it directly. We circumvent this difficulty by using the radius defined for each conditional gaussian of the prior, ignoring interactions between each layer. The pseudocode for optimization in the latent space is shown in Algorithm 2. We project each layer of latent variables separately onto the radius defined by their conditional gaussian distribution, and then optimize with respect to the $N-1$ angles.

To further ensure that the optimization is constrained to well-trained regions of latent space, we add $\beta \cdot \log p(z)$ to the objective function, where $\beta$ is a hyperparameter. Finally, to moderate the strictly monotonic nature of linear regressors, we apply an element-wise hard tanh to all latent variables before the regressor, with a linear region that encompasses all values observed in the training set.

To compare with previous work as fairly as possible, we optimize 1000 random samples from the prior to convergence, collecting the last point from each trajectory with a valid SMILES decoding.

---

**Algorithm 1:** Initialize Angles

---

**output :** Angular coordinates of latent variable sample on the spherical shell with radius $\sqrt{N-1}$

**for** $i \leftarrow 0$ **to** $K$ **do**                          // For each layer $i$ in the hierarchy
  $\quad \epsilon_i \leftarrow N(0, I)$
  $\quad \hat{\epsilon}_i \leftarrow \frac{\epsilon_i}{||\epsilon_i||} \cdot \sqrt{N-1}$ ;                          // project onto spherical shell
  $\quad \theta_i \leftarrow \texttt{ToPolarCoords}(\hat{\epsilon}_i)$
**end**
**return** $\{\theta_i\}_1^N$

---

---

**Algorithm 2:** Optimization in latent space with hierarchical radius constraint

---

**input  :** Property models: $[f_1, \ldots, f_m]$, Prior distribution:
  $\quad [p(z_K|NN_K(z_{<K})), \ldots, p(z_1|NN_0(z_0)), p(z_0)]$, Objective function: $O(\cdot)$
**output :** Spherical coordinates of a molecule in latent space with converged property values
initialize $\{\theta_i\}_1^K \leftarrow \texttt{InitializeAngles()}$ ;
// The first layer of the prior is a standard gaussian
$\mu_0 \leftarrow 0, \sigma_0 \leftarrow 1$;
**for** $i \leftarrow 0$ **to** $K$ **do**                          // For each layer $i$ in the hierarchy
  $\quad z_i \leftarrow \texttt{ToCartesianCoords(}\theta_i\texttt{)}$
  $\quad \hat{z}_i \leftarrow z_i \cdot \sigma_i + \mu_i$ ;                    // Re-parametrize standard gaussian variable to
  $\quad$ conditional gaussian at position $i$ in the hierarchy
  $\quad . \mu_{i+1}, \sigma_{i+1} \leftarrow \texttt{NN}_i(\hat{z}_{<i})$ ;                    // Compute $\mu$ and $\sigma$ of the next level
**end**
// Optimize $\{\theta_i\}$ until the objective function $O(\cdot)$ has converged
$\{\theta_i^*\}_1^N \leftarrow \texttt{GradientDescent(}O(\{f_j\}_1^M, \{\hat{z}_i(\theta_i)\}_1^K)\texttt{)}$ ;
**return** $\{\theta_i^*\}_1^K$

---

From these 1000 points, we evaluate the true molecular property on the 100 points for which the predicted property value is largest. Of these 100 values, we report the three largest. However, optimization within our latent space is computationally inexpensive, and requires no additional property measurement data. We could somewhat improve molecular optimization at minimal expense by constructing additional optimization trajectories in latent space, and evaluating the true molecular properties on the best points from this larger set.

Molecular optimization is quite robust to hyperparameters. We considered ADAM learning rates in $\{0.1, 0.01, 0.001, 0.0001\}$ and $\beta \in \{0.1, 0.01, 0.001, 0.0001\}$.

### B.4    SUMMARY OF NOVEL CONTRIBUTIONS

Starting with the work of Gómez-Bombarelli et al. (2018), previous molecular variational autoencoders have used one consistent SMILES string as both the input to the RNN encoder and the target of the RNN decoder. Any single SMILES string explicitly represents only a subset of the pathways in the molecular graph. Correspondingly, the recurrent neural networks in these encoders implicitly propagated information through only a fraction of the possible pathways. Kipf & Welling (2016b), Liu et al. (2018), and Simonovsky & Komodakis (2018), amongst others, trained molecular VAEs with graph convolutional encoders, which pass information through all graph pathways in parallel, but at considerable computational expense. None of these works used enough layers of graph convolutions to transfer information across the diameter of the average molecule in standard drug design datasets. This is partially overcome by Lusci et al. (2013) who ensemble RNN-based representations of multiple directed-acyclic graphs of a single molecule for property prediction. The All SMILES VAE introduces the use of multiple SMILES strings of a single, common molecule as input to a RNN encoder, with pooling of homologous messages amongst the hidden representations associated with different SMILES strings. This allows information to flow through all pathways of the molecular graph, but can efficiently propagate information across the entire width of the molecule in a single layer.

Bjerrum & Sattarov (2018) and Winter et al. (2019a) trained sequence-to-sequence transcoders to map between different SMILES strings of the same molecule. These transcoders do not define an explicit generative model over molecules, and their latent spaces have no prior distributions. The All SMILES VAE extends this approach to variational autoencoders, and thereby learns a SMILES-derived generative model of molecules, rather than SMILES strings. The powerful, learned, hierarchical prior of the All SMILES VAE regularizes molecular optimization and property prediction. To ensure that molecular property optimization searches within the practical support of the prior, containing almost all of its probability mass, we introduce a hierarchical radius constraint on optimization with respect to the latent space.

## C    EXTENDED RESULTS

We compare the performance of the All SMILES VAE to a variety of state-of-the-art algorithms that have been evaluated on standard molecular property prediction and optimization tasks. In particular, we compare to previously published results on the character/chemical VAE (CVAE) (Gómez-Bombarelli et al., 2018) (with results reported in (Kusner et al., 2017)), grammar VAE (GVAE) (Kusner et al., 2017), syntax-directed VAE (SD-VAE) (Dai et al., 2018), junction tree VAE (JT-VAE) (Jin et al., 2018), NeVAE (Samanta et al., 2018), semisupervised VAE (SSVAE) (Kang & Cho, 2018), graph convolutional policy network (GCPN) (You et al., 2018), molecule deep Q-network (MolDQN) (Zhou et al., 2018), and the DeepChem (Wu et al., 2018) implementation of extended connectivity fingerprints (ECFP) (Rogers & Hahn, 2010) and graph convolutions (GraphConv) (Duvenaud et al., 2015; Kearns et al., 2016; Wu et al., 2018). Extended connectivity fingerprints are a fixed-length hash of local fragments of the molecule, used as input to conventional machine learning techniques such as random forests, support vector machines, and non-convolutional neural networks (Wu et al., 2018). For toxicity prediction, we also compare to PotentialNet (Feinberg et al., 2018), ToxicBlend (Zaslavskiy et al., 2019), and the results of (Li et al., 2017).

### C.1    RECONSTRUCTION ACCURACY AND VALIDITY

Previous molecular variational autoencoders have been evaluated using the percentage of molecules that are correctly reconstructed when sampling from both the approximating posterior $q(z|x)$ and the conditional likelihood $p(x|z)$ (reconstruction accuracy), and the percentage of samples from the prior $p(z)$ and conditional likelihood $p(x|z)$ that are valid SMILES strings (validity). While these measure have intuitive appeal, they reflect neither the explicit training objective (the ELBO), nor the requirements of molecular optimization. In particular, when optimizing molecules via the latent space, a deterministic decoder ensures that each point in latent space is associated with a single set of well-defined molecular properties.

The All SMILES VAE is trained on a more difficult task than previous molecular VAEs, since the reconstruction targets are different SMILES encodings than those input to the approximating posterior. This ensures that the latent representation only captures the molecule, rather than its particular SMILES encoding, but it requires the decoder LSTM to produce a complex, highly multimodal distribution over SMILES strings. As a result, samples from the decoder distribution are less likely to correspond to the input to the encoder, either due to syntactic or semantic errors.

To compensate for this unusually difficult decoding task, we evaluate the All SMILES VAE using a beam search over the decoder distribution.[6] That is, we decode to the single SMILES string estimated to be most probable under the conditional likelihood $p(x|z)$. This has the added benefit of defining an unambiguous decoding for every point in the latent space, simplifying the interpretation of optimization in the latent space (Section 4.2). However, it renders the reconstruction and validity reported in Section 4 incommensurable with much prior work, which use stochastic encoders and decoders.

### C.2    PROPERTY PREDICTION

Rather than jointly modeling the space of molecules and their properties, some earlier molecular variational autoencoders first trained an unsupervised VAE on molecules, extracted their latent

---

[6]The full decoder distribution is still used for training.

representations, and then trained a sparse Gaussian process over molecular properties as a function of these fixed latent representations (Dai et al., 2018; Jin et al., 2018; Kusner et al., 2017; Samanta et al., 2018). Sparse Gaussian processes are parametric regressors, with the location and value of the inducing points trained based upon the entire supervised dataset (Snelson & Ghahramani, 2006). They have significantly more parameters, and are corresponding more powerful, than linear regressors.

Molecular properties are only a smooth function of the VAE latent space when the property regressor is trained jointly with the generative model (Gómez-Bombarelli et al., 2018). Results using a sparse Gaussian process on the latent space of an unsupervised VAE are very poor compared to less powerful regressors trained jointly with the VAE. Our property prediction is two orders of magnitude more accurate than sparse Gaussian process regression on an unsupervised VAE latent representation, as shown in Table 4.

Table 4: Root-mean-square error of the log octanol-water partition coefficient (logP) on the ZINC250k dataset. Results other than the All SMILES VAE are those reported in the cited papers.

| MODEL | RMSE |
|---|---|
| CHARACTER VAE (CVAE) (GÓMEZ-BOMBARELLI ET AL., 2018; KUSNER ET AL., 2017) | 1.504 |
| GRAMMAR VAE (GVAE) (KUSNER ET AL., 2017) | 1.404 |
| SYNTAX-DIRECTED VAE (SD-VAE) (DAI ET AL., 2018) | 1.366 |
| JUNCTION TREE VAE (JT-VAE) (JIN ET AL., 2018) | 1.290 |
| NEVAE (SAMANTA ET AL., 2018) | 1.23 |
| **All SMILES** | **0.011 ± 0.001** |

We report numerical results on semi-supervised property prediction, as well as comparisons from (Kang & Cho, 2018), in Table 5. Our mean absolute error is at least three times smaller than comparison algorithms on the log octanol-water partition coefficient (logP) and molecular weight (MW).

Table 5: Mean absolute error (MAE) of semi-supervised property prediction on the log octanol-water partition coefficient (logP), molecular weight (MW), and the quantitative estimate of drug-likeness (QED) on ZINC310k dataset. Results other than the All SMILES VAE are those reported by (Kang & Cho, 2018).

| MODEL | % LABELED | MAE LOGP | MAE MW | MAE QED |
|---|---|---|---|---|
| ECFP | 50% | 0.180 ± 0.003 | 9.012 ± 0.184 | 0.023 ± 0.000 |
| GRAPHCONV | 50% | 0.086 ± 0.012 | 4.506 ± 0.279 | 0.018 ± 0.001 |
| SSVAE | 50% | 0.047 ± 0.003 | 1.05 ± 0.164 | 0.01 ± 0.001 |
| ALL SMILES | 50% | 0.007 ± 0.002 | 0.21 ± 0.07 | 0.0064 ± 0.0002 |
| ECFP | 20% | 0.249 ± 0.004 | 12.047 ± 0.168 | 0.033 ± 0.001 |
| GRAPHCONV | 20% | 0.112 ± 0.015 | 4.597 ± 0.419 | 0.021 ± 0.002 |
| SSVAE | 20% | 0.071 ± 0.007 | 1.008 ± 0.370 | 0.016 ± 0.001 |
| ALL SMILES | 20% | 0.009 ± 0.002 | 0.33 ± 0.06 | 0.0079 ± 0.0003 |
| ECFP | 10% | 0.335 ± 0.005 | 15.057 ± 0.358 | 0.045 ± 0.001 |
| GRAPHCONV | 10% | 0.148 ± 0.016 | 5.255 ± 0.767 | 0.028 ± 0.003 |
| SSVAE | 10% | 0.090 ± 0.004 | 1.444 ± 0.618 | 0.021 ± 0.001 |
| ALL SMILES | 10% | 0.014 ± 0.002 | 0.30 ± 0.06 | 0.0126 ± 0.0006 |
| ECFP | 5% | 0.380 ± 0.009 | 17.713 ± 0.396 | 0.053 ± 0.001 |
| GRAPHCONV | 5% | 0.187 ± 0.015 | 6.723 ± 2.116 | 0.034 ± 0.004 |
| SSVAE | 5% | 0.120 ± 0.006 | 1.639 ± 0.577 | 0.028 ± 0.001 |
| ALL SMILES | 5% | 0.036 ± 0.004 | 0.4 ± 0.1 | 0.0217 ± 0.0003 |

As a visual demonstration of the accuracy of property prediction, in Figure 12 we show the predicted logP of a 2D slice of latent space subject to the hierarchical radius constraint, alongside the true logP of the molecules decoded from this slice (identical to Figure 6a).

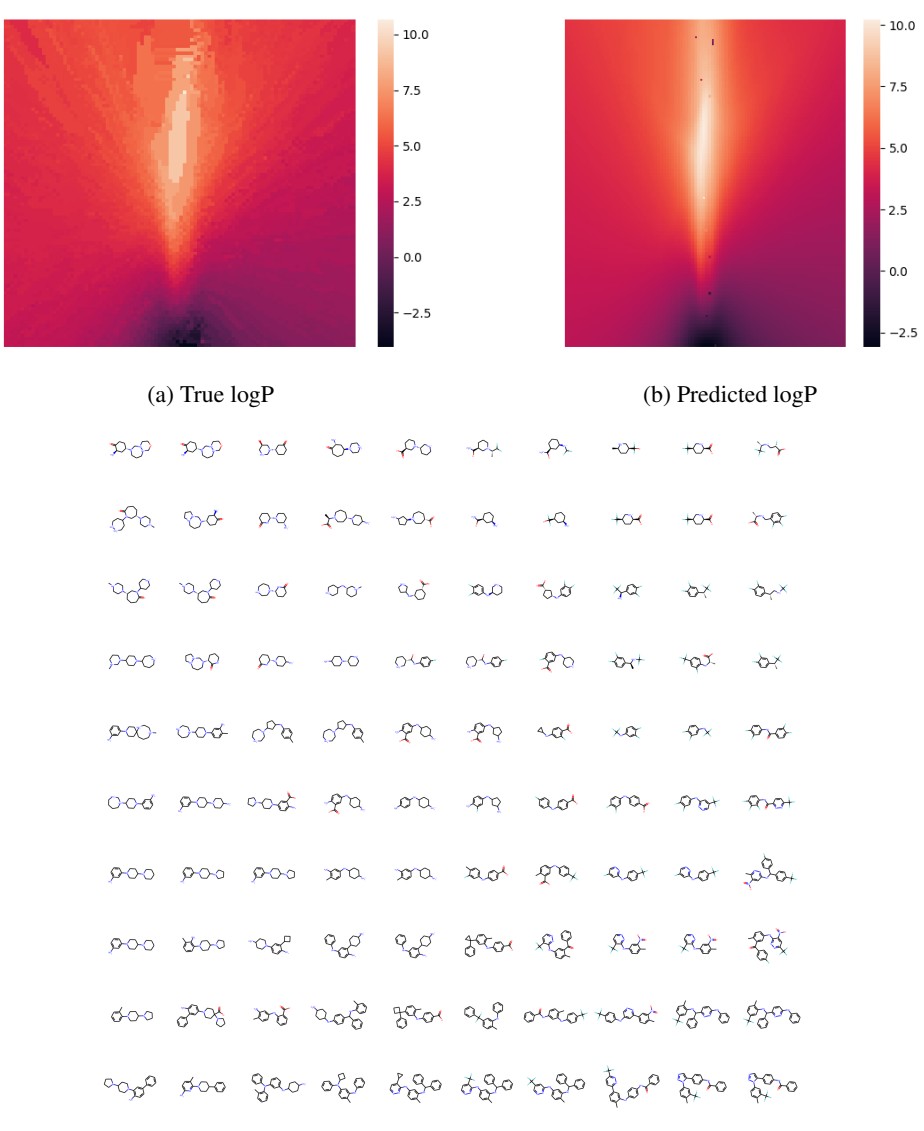

(a) True logP                    (b) Predicted logP

(c) Sheet of molecules

Figure 12: Dense decodings of true logP (a) and predicted logP (b) along a local 2D sheet in latent space, with the y axis aligned with the trained logP regressor. We also display a coarse sampling of the molecules corresponding to the logP heatmap (c).

Pathways on which activity (active or inactive) is assessed for the Tox21 dataset include seven nuclear receptor signaling pathways: androgen receptor, full (NR-AR) androgen receptor, LBD (NR-AR-LBD); aryl hydrocarbon receptor (NR-AHR); aromatase (NR-AROMATASE); estrogen receptor alpha, LBD (NR-ER-LBD); estrogen receptor alpha, full (NR-ER); and peroxisome proliferator-activated receptor gamma (NR-PPAR-GAMMA). The Tox21 dataset also includes activity assessments for five stress response pathways: nuclear factor (erythroid-derived 2)-like 2/antioxidant responsive element (SR-ARE); ATAD5 (SR-ATAD5); heat shock factor response element (SR-HSE); mitochondrial membrane potential (SR-MMP); and p53 (SR-p53). We report the area under the receiver operating characteristic curve (AUC-ROC) on each assay independently in Table 6. The average of these AUC-ROCs is reported in Table 1. We do not include the result of (Kearnes et al.,

2016) in Table 1, since it is not evaluated on the same train/validation/test split of the Tox21 dataset, and so is not commensurable.

Table 6: Area under the receiver operating characteristic curve (AUC-ROC) per assay on the Tox21 dataset.

| NR-AR | NR-AR-LBD | NR-AHR | NR-AROMATASE | NR-ER | NR-ER-LBD |
|---|---|---|---|---|---|
| 0.864 | 0.921 | 0.909 | 0.908 | 0.719 | 0.811 |
| NR-PPAR-GAMMA | SR-ARE | SR-ATAD5 | SR-HSE | SR-MMP | SR-p53 |
| 0.935 | 0.860 | 0.870 | 0.901 | 0.927 | 0.882 |

### C.3    MOLECULAR OPTIMIZATION

We present an optimization trajectory for the quantitative estimate of drug-likeness (QED) in Figure 13. For the molecules depicted in Figure 7, we scaled $D_{KL}(q(z|x)||p(z)))$ in the ELBO (Equation 1) of the All SMILES VAE by the number of SMILES strings in the decoder. This renders the loss function analogous to that of many parallel single-SMILES VAEs, but with message passing between encoders leading to a shared latent representation. If we leave the KL term unscaled, latent space embeddings are subject to less regularization forcing them to match the prior distribution. Optimization of molecular properties with respect to the latent space therefore searches over a wider space of molecules, which are less similar to the training set. In Figure 14, we show that such an optimization for penalized log P finds very long aliphatic chains, with penalized log P values as large as 42.46.

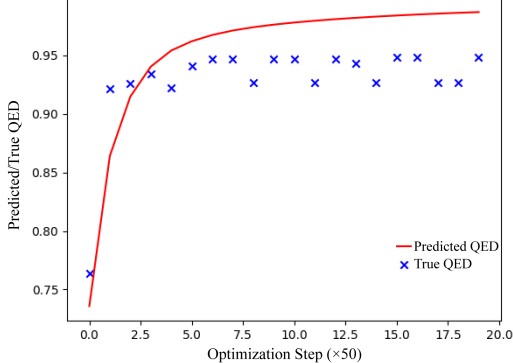

Figure 13: Predicted (red line) and true (blue x's) quantitative estimate of drug-likeness (QED) over the optimization trajectory resulting in the molecule with the maximum observed true QED (0.948).

```
CCCCCCCCCCCCCCCCCCCCCCCCCCCCC       CCCCCCCCCCCCCCCCCCCCCCCCCCCCC       CCCCCCCCCCCCCCCCCCCCCCCCCCCCCC
CCCCCCCCCCCCCCCCCCCCCCCCCCCCC       CCCCCCCCCCCCCCCCCCCCCCCCCCCCC       CCCCCCCCCCCCCCCCCCCCCCCCCCCCCC
CCCCCCCCCCCCCCCCCCCCCCCCCCCCC       CCCCCCCCCCCCCCCCCCCCCCCCCCCCC       CCCCCCCCCCCCCCCCCCCCCCCCCCCCCC
CCCCCCCCCOCCCCCCCCCCCCCCCCCCC       CCCCCCCCCCCCCCCCCCCCCCCCCCCCC       CCCCCCCCOCOCCCCCCCCCCCCCCCCCCC
CCCOC                               OCCOC                               CCCOC
```

Figure 14: Molecules with the top three true penalized LogP values produced by gradient-based optimization subject to the hierarchical radius constraint in the All SMILES VAE, but with the KL term unscaled by the number of SMILES strings in the decoder. Molecules are shown as SMILES strings, wrapped across multiple lines, as they are too large to be properly rendered into an image.

### C.4    ABLATION OF HIERARCHICAL RADIUS CONSTRAINT

Table 7 shows that the hierarchical radius constraint significantly improves molecular optimization. In contrast to Table 2, optimization is performed on penalized logP alone, without a log prior regularizer.

This produces better results without the radius constraint, and so constitutes a more conservative ablation experiment.

Table 7: Effect of the hierarchical radius constraint on penalized logP optimization. Predicted penalized logP was evaluated on 1000 optimization trajectories. From these, the true logP was evaluated on the 100 best trajectories, and the top three true penalized logPs are reported. Each optimization was repeated 5 times.

| ABLATION | 1ST BEST LOGP | 2ND BEST LOGP | 3RD BEST LOGP |
|---|---|---|---|
| WITH RADIUS CONSTRAINT | $17.0 \pm 3.0$ | $16.0 \pm 2.0$ | $14.8 \pm 0.3$ |
| WITHOUT RADIUS CONSTRAINT | $8.5044 \pm 0.0$ | $6.9526 \pm 0$ | $5.36 \pm 0.05$ |

## D   SMILES GRAMMAR CAN BE ENFORCED WITH A PUSHDOWN AUTOMATON

The subset of the SMILES grammar (Weininger, 1988) captured by (Dai et al., 2018) and (Kusner et al., 2017) is equivalent to the context-free grammar shown in Figure 15. This subset does not include the ability to represent multiple disconnected molecules in a single SMILES string, multiple fragments that are only connected by ringbonds, or wildcard atoms. element_symbols includes symbols for every element in the periodic table, including the aliphatic_organic symbols.

$$
\begin{aligned}
\text{chain} &\rightarrow \text{branched\_atom rest\_of\_chain} \\
\text{rest\_of\_chain} &\rightarrow \epsilon \text{ — bond? chain} \\
\text{bond} &\rightarrow \text{'-' — '=' — '\#' — '\$' — ':' — '/' — '\backslash'} \\
\text{branched\_atom} &\rightarrow \text{atom ringbond* branch*} \\
\text{ringbond} &\rightarrow \text{bond digit? digit} \\
\text{branch} &\rightarrow \text{'(' bond? chain ')'} \\
\text{atom} &\rightarrow \text{aliphatic\_organic — aromatic\_organic — bracket\_atom} \\
\text{aliphatic\_organic} &\rightarrow \text{'B' — 'C' — 'N' — 'O' — 'S' — 'P' — 'F' — 'Cl' — 'Br' — 'I'} \\
\text{aromatic\_organic} &\rightarrow \text{'b' — 'c' — 'n' — 'o' — 's' — 'p'} \\
\text{bracket\_atom} &\rightarrow \text{'[' isotope? symbol chiral? hcount? charge? class? ']'} \\
\text{isotope} &\rightarrow \text{digit? digit? digit} \\
\text{symbol} &\rightarrow \text{element\_symbols — aromatic\_symbols} \\
\text{aromatic\_symbols} &\rightarrow \text{'c' — 'n' — 'o' — 'p' — 's' — 'se' — 'as'} \\
\text{chiral} &\rightarrow \text{'@' — '@@' — '@TH1' — '@TH2' — '@AL1' — '@AL2' —} \\
&\qquad \text{'@SP1' — '@SP2' — '@SP3' — '@TB1' — '@TB2' } \cdots \text{ '@TB30' —} \\
&\qquad \text{'@OH1' — '@OH2' } \cdots \text{ '@OH30'} \\
\text{hcount} &\rightarrow \text{'H' digit?} \\
\text{charge} &\rightarrow \text{'-' digit? — '+' digit?} \\
\text{class} &\rightarrow \text{':' digit? digit? digit?} \\
\text{digit} &\rightarrow \text{'0' — '1' — '2' — '3' — '4' — '5' — '6' — '7' — '8' — '9'}
\end{aligned}
$$

Figure 15: Context-free grammar of SMILES strings

Productions generally begin with a unique, defining symbol or set of symbols. Exceptions include bond and charge (both can begin with −), and aromatic_organic and aromatic_symbols (both include c, n, o, s, and p), but these pairs of productions never occur in the same context, and so cannot be confused. The particular production for chiral can only be resolved by parsing characters up to the next production, but the end of chiral and the identity of the subsequent production can be inferred from its first symbol of the production after chiral. Alternatively, the strings of chiral can be encoded as monolithic tokens.

Whenever there is a choice between productions, the true production is uniquely identified by the next symbols. The only aspect of the SMILES grammar that requires more than a few bits of memory is the matching of parentheses, which can be performed in a straightforward manner with a pushdown automaton. As a result, parse trees (Dai et al., 2018; Kusner et al., 2017) need not be explicitly constructed by the decoder to enforce the syntactic restrictions of SMILES strings. Rather, the SMILES grammar can be enforced with a pushdown automaton running in parallel with the decoder RNN. The state of the pushdown automaton tracks progress within the representation of each atom, and the sequence of atoms and bonds. The set output symbols available to the decoder RNN is restricted to those consistent with the current state of the pushdown automaton. ( and [ are pushed onto the stack when are emitted, and must be popped from the top of the stack in order to emit ) or ] respectively.

For example, in addition to simple aliphatic organic (`B`, `C`, `N`, `O`, `S`, `P`, `F`, `Cl`, `Br`, or `I`) or aromatic organic (`b`, `c`, `n`, `o`, `s`, or `p`) symbols, an `atom` may be represented by a pair of brackets (requiring parentheses matching) containing a sequence of isotope number, atom symbol, chiral symbol, hydrogen count, charge, and class. With the exception of the atom symbol, each element of the sequence is optional, but is easily parsed by a finite state machine. `isotope`, `symbol`, `chiral`, `hcount`, `charge`, and `class` can all be distinguished based upon their first character, so the position in the progression can be inferred trivially.[7]

When parsing `branched_atom`, all productions after the initial `atom` are `ringbonds` until the first `(`, which indicates the beginning of a `branch`. After observing a `)`, and popping the complementary `(` off of the stack, the SMILES string is necessarily in the third component of a `branched_atom`, since only a `branched_atom` can emit a `branch`, and only `branch` produces the symbol `)`. The next symbol must be a `(`, indicating the beginning of another `branch`, or one of the first symbols of `rest_of_chain`, since this must follow the `branched_atom` in the `chain` production.

### D.1  RINGBOND AND VALENCE SHELL SEMANTIC CONSTRAINTS

Similarly, the semantic restrictions of ringbond matching and valence shell constraints can be enforced during feedforward production of a SMILES string using a pushdown stack and a small (100-element) random access memory. Our approach depends upon the presence of matching bond labels at both sides of a ringbond, which is allowed but not required in standard SMILES syntax. We assume the trivial extention of the SMILES grammar to include this property.

`ringbond`s are constrained to come in pairs, with the same bond label on both sides. Whenever a given `ringbond` is observed, flip a bit in the random access memory corresponding to the ring number (the set of digits after the `bond`). When the `ringbond` bit is flipped on, record the associated `bond` in the random access memory associated with the ring number; when the `ringbond` bit is flipped off, require that the new `bond` matches the recorded `bond`, and clear the random access memory of the `bond`. The molecule is only allowed to terminate (`rest_of_chain` produces $\epsilon$ rather than `bond? chain`) when all `ringbond` bits are off (parity is even). The decoder may receive as input which `ringbond`s are open, and the associated `bond` type, so it can preferentially close them.

The set of nested atomic contexts induced by `chain`, `branched_atom`, and `branch` can be arbitrarily deep, corresponding to the depth of branching in the spanning tree realized by a SMILES string. As a result, the set of SMILES symbols describing bonds to a single atom can be arbitrarily far away from t=he associated `atom`. However, once a branch is entered, it must be traversed in its entirety before the SMILES string can return to the parent atom. For each atom, it is sufficient to push the valence shell information onto the stack as it is encountered. If the SMILES string enters a branch while processing an atom, simply push on a new context, with a new associated root atom. Once the branch is completed, pop this context off the stack, and return to the original atom.

More specifically, each atom in the molecule is completely described by a single `branched_atom` and the `bond` preceding it (from the `rest_of_chain` that produced the `branched_atom`). Within each successive pair of `bond` and `branched_atom`, track the sum of the incoming `rest_of_chainbond`, the internal `ringbond` and `branch` bonds, and outgoing

---

[7] `symbol` and `hcount` can both start with 'H', but symbol is mandatory, so there is no ambiguity.

`rest_of_chain` bond (from the succeeding `rest_of_chain`) on the stack. That is, each time a new bond is observed from the atom, pop off the old valence shell count and push on the updated count. Require that the total be less than a bound set by the atom; any remaining bonds are filled by implicity hydrogen atoms. Provide the number of available bonds as input to the decoder RNN, and mask additional `ringbonds` and `branches` once the number of remaining available bonds reaches one (if there are still open `ringbonds`) or zero (if all `ringbonds` are closed). Mask the outgoing `bond`, or require that `rest_of_chain` produce $\epsilon$, based upon the number of remaining available bonds.

### D.2 REDUNDANCY IN GRAPH-BASED AND SMILES REPRESENTATIONS OF MOLECULES

To avoid the degeneracy of SMILES strings, for which there are many encodings of each molecule, some authors have advocated the use of graph-based representations (Li et al., 2018; Liu et al., 2018; Ma et al., 2018; Simonovsky & Komodakis, 2018). While graph-based processing may produce a unique representation in the encoder, it is not possible to avoid degeneracy in the decoder. Parse trees (Dai et al., 2018; Kusner et al., 2017), junction trees (Jin et al., 2018), lists of nodes and edges (Li et al., 2018; Liu et al., 2018; Samanta et al., 2018), and vectors/matrices of node/edge labels (De Cao & Kipf, 2018; Ma et al., 2018; Simonovsky & Komodakis, 2018) all imply an ordering amongst the nodes and edges, with many orderings describing the same graph. Canonical orderings can be defined, but unless they are obvious to the decoder, they make generative modeling harder rather than easier, since the decoder must learn the canonical ordering rules. Graph matching procedures can ensure that probability within a generative model is assigned to the correct molecule, regardless of the order produced by the decoder (Simonovsky & Komodakis, 2018). However, they do not eliminate the degeneracy in the decoder's output, and the generative loss function remains highly multimodal.

