# OpenReview forum: "All SMILES Variational Autoencoder for Molecular Property Prediction and Optimization"
_ICLR.cc/2020/Conference — Reject_

### Official Review · AnonReviewer1 · 2019-10-23
**Official Blind Review #1**

**Rating:** 3

**Review:**

This paper presents a VAE-based method to predict molecular properties as well as to design a novel molecular architecture.
The method employs as a basic input the SMILES representation of molecules which are not well defined in terms of the representation, though.
To tackle the issue, the authors formulate the method based on (1) multiple inputs of SMILES strings, (2) the character-wise feature fusion across those multiple strings and (3) network training by multiple output targets of SMILES strings different from the input ones.
As a result, the method provides the fixed-length latent representation which is robust against the variations of the SMILES representation and is useful for predicting the molecular properties and optimizing the molecular design.
The experimental results on the tasks of the property prediction and molecular optimization using the benchmark datasets demonstrate the effectiveness of the proposed method in comparison with the others.

Though this paper provides some contributions to the field of molecular graph representation, it contains flaws in presentation, lacking details of technical contents; thus, the paper is regarded as "borderline". I would like the authors to properly provide the details in the following points.

* This paper lacks some important technical contents, making it hard to understand.
- What is the actual form of the property predictor used in p(\rho|z). And, in the first place of the paper, it would be better to explain what kind of and how many properties of the molecular are considered.
- How is the decoder constructed and trained? Though the authors state that a single-layer LSTM is employed as the decoder, there is no clear description how to cope with the multiple outputs (decoder targets) in training the LSTM.
- What does \theta in p_\theta(z) mean? In the VAE, p(z) works as a simple prior on z.
- The description, especially in Sec. 3 for the proposed method, is rather poorly presented by using less amount of math. At least, the authors should first depict the overall architecture of the method through clarifying the above-mentioned technical points.
- It is unclear how to apply the proposed method to the semi-supervised learning framework? Is it re-formulated in the SS-VAE [a] framework?
- The comparison experiments seem to be inconsistent. The proposed method is compared with different methods in different datasets/tasks. Toward fair comparison, it should be evaluated in comparison with some baselines including such as JT-VAE consistently.

[a] Kingma, D. P.; Mohamed, S.; Rezende, D. J.; Welling, M. Semi-Supervised Learning with Deep Generative Models. Proceedings of the 28th Annual Conference on Neural Information Processing Systems. 2014; pp 3581–3589.

* The authors cope with the fluctuations regarding the SMILES representation by means of RNN. There, however, are some approaches to canonicalize the SMILES representation itself such as by canonical SMILES. The authors have to mention those other approaches and discuss the superiority of the method over them; hopefully the proposed method should be compared with the naive one simply combining VAE [Kang&Cho18] and canonical SMILES representation. And to further understand the difficulty in SMILES variations, it would be better to show the averaged cardinality of SMILES representation per molecular.


Minor comment:
* In aggregating the RNN features for each character in SMILES strings, it might be possible to incorporate some structural knowledge into the process; for example, "C"s in benzene share the identical feature representation through fusing all those features.

**Experience Assessment:**

I have read many papers in this area.

**Review Assessment: Checking Correctness Of Derivations And Theory:**

I assessed the sensibility of the derivations and theory.

**Review Assessment: Checking Correctness Of Experiments:**

I assessed the sensibility of the experiments.

**Review Assessment: Thoroughness In Paper Reading:**

I read the paper at least twice and used my best judgement in assessing the paper.

---

> ### Author Response · Authors · 2019-11-12
> **Response part 1**
>
> Comment 1: - What is the actual form of the property predictor used in p(rho|z). And, in the first place of the paper, it would be better to explain what kind of and how many properties of the molecular are considered.
>
> Response 1:
> In general, p(rho|z) is a simple linear transformation of the latent variables into a scalar, followed by an activation function that transforms the property prediction into the correct range. QED is in the range [0,1] and Tox21 is binary, so we use a sigmoid activation function; penalized logP is in R (the real numbers), so we use the identity activation function (no activation function); and molecular weight is in R^+ (non-negative real numbers), so we use a softplus activation function. This is described in Section 4.1 of the text, but we have added additional explanation in Section 3 to enhance the clarity.
>
> Models are trained on one type of property: either QED, penalized logP, molecular weight, or all twelve binary Tox21 toxicity categories. For instance, in Figure 5a we only train on logP; in Figure 5b, we train a separate instance of our model on molecular weight; in Table 1b, we train an independent instance of our model on Tox21; etc.
>
> Comment 2:
>  How is the decoder constructed and trained? Though the authors state that a single-layer LSTM is employed as the decoder, there is no clear description how to cope with the multiple outputs (decoder targets) in training the LSTM.
>
> Response 2:
> The decoder is trained using teacher forcing on a set of n SMILES strings of a single molecule in parallel, all using the same latent representation. This is analogous to training using minibatch of molecules, but all n SMILES strings of a single, common molecule are decoded from the same latent representation. The total decoder log-likelihood is the sum of the log-likelihoods on each of these SMILES strings.
>
> The decoder is trained using teacher forcing to be able to produce all SMILES strings of a target molecule. When the LSTM starts decoding, it will be uncertain as to which SMILES is being decoded. For each successive character of the target SMILES string, the log-probability assigned by the decoder LSTM to the correct character is evaluated, and the correct character is fed into the decoder LSTM. As the LSTM sees more of the previous true outputs, its predictions as to which SMILES string of the molecule it needs to decode becomes more accurate.
>
> This is described in Section 3 of the decoder architecture description, but we have edited the text for increased clarity.
>
> Comment 3:
> What does theta in p_{theta}(z) mean? In the VAE, p(z) works as a simple prior on z.
>
> Response 3:
> In the VAE literature, work has been done in parametrizing prior distributions to make them more expressive than the standard Gaussian used in the vanilla VAE [Rolfe 2016; Chen et al., 2016; Tomczak, 2017]. The theta in p_{theta}(z) denotes the parameters of such parametrized priors. We describe the architecture of the parametrized prior in Section 3, but have edited the text for increased clarity. Briefly, we use a hierarchical prior, where each autoregressive conditional Gaussian distribution is parametrized by a two layer neural network conditioned on the concatenated latent variables sampled from the previous layer. With mathematical notation : p_{\theta}(z_1, z_2, z_3, z_4) = p(z_1) \cdot \prod^4_{i=2} p_{\theta_i}(z_i|z_{<i})  where \theta_i represents the parameters of the two layer neural network which takes z_{<i} as input, and p(z_1) is just the unparameterized standard Gaussian.

---

> ### Author Response · Authors · 2019-11-12
> **Response part 2**
>
> Comment 4:
>  It is unclear how to apply the proposed method to the semi-supervised learning framework? Is it re-formulated in the SS-VAE
>
> Response 4:
> Rather than a semi-supervised VAE [Kingma et al., 2014] applied to SMILES strings by [Kang and Cho, 2018], we use a generative model over both SMILES strings and properties, for which the encoder is only provided SMILES strings (disjoint from the targets of the decoder). That is, the encoder q(z|x) is provided only SMILES strings of the molecule, whereas the decoder p(x,rho|z) = p(x|z) \cdot p(rho|z) is required to reconstruct both the (disjoint) SMILES strings and the molecular properties from the latent representation. Independent networks p(x|z) and p(rho|z) decode from the latent representation to the SMILES strings (an LSTM) and the molecular properties (a linear transformation followed by a scalar activation function), and the joint conditional distribution over SMILES strings and molecular properties given the latent representation is just the product of the outputs of these independent decoder networks. To train on semi-supervised data, the decoder is only trained to reconstruct the molecular properties on those elements of the dataset for which this information is available. For the unsupervised elements of the dataset, the conditional log-likelihood of the molecular properties is removed, or equivalently multiplied by zero. We have clarified this architecture in the paper.
>
> In contrast, [Kang and Cho, 2018] predict the molecular properties from the molecule directly in the approximating posterior q(rho|x), then predict latent variables from the property q(z|x,rho). They also condition the decoder p(x|rho,z) on the property directly.
>
> Comment 5:
>
> The comparison experiments seem to be inconsistent. The proposed method is compared with different methods in different datasets/tasks. Toward fair comparison, it should be evaluated in comparison with some baselines including such as JT-VAE consistently.
>
> Response 5:
>
> To demonstrate the flexibility of our approach, we evaluate on multiple tasks, including supervised and semi-supervised learning, property optimization, and generative modelling. In all cases, we compare to the current state-of-the-art on each task. This ensures that the comparison methods are carefully tuned for the target task, and do not underestimate their performance due to suboptimal hyperparameter selection.
>
> In cases such as ToxicBlend, the state-of-the-art for one task is not structurally suitable for other tasks, and so it is not possible to apply some algorithms across all tasks. In the case of JT-VAE [Jin et al., 2018], we demonstrate superior performance on penalized logP and QED molecular property optimization tasks (Table 2), supervised learning (Table 4 of Appendix C).
>
>
> Comment 6:
> The authors cope with the fluctuations regarding the SMILES representation by means of RNN. There, however, are some approaches to canonicalize the SMILES representation itself such as by canonical SMILES. The authors have to mention those other approaches and discuss the superiority of the method over them …
>
> Response 6:
>
> While canonical SMILES strings associate a unique SMILES string with each molecule, this string has no privileged semantics. It effectively uses a hash function to choose a consistent but otherwise arbitrary SMILES string for each molecular graph. Generative models of canonical SMILES strings learn a model of SMILES strings, rather than molecules. Similar molecules may have very different canonical SMILES strings, and so dissimilar latent representations. In contrast, by using a disjoint set of SMILES strings of a single molecule as encoder input and decoder target, we train a generative model over molecules rather than individual SMILES strings, facilitating the use of nearby latent representations for similar molecules.
>
> We explore the impact of our novel multiple-SMILES approach versus a series of more conventional representations in our ablation experiments. In ONE SMILES ENC and ONE SMILES ENC/DEC (!=), we successively remove multiple-SMILES input to the encoder and decoder, but still randomly choose SMILES strings from the full set of SMILES strings capturing the molecular graph. Finally, in ONE SMILES ENC/DEC (=), we use only the canonical SMILES string of the molecular graph. As can be seen in Table 3, all of these variations hurt performance, with the canonical SMILES representation yielding the worst performance of all. We also investigate the performance of a vanilla VAE with a standard Gaussian prior and factorial approximating posterior in the NO POSTERIOR HIERARCHY experiment of Table 3, demonstrating a similar reduction in performance relative to our hierarchical approximating posterior and prior.

---

> ### Author Response · Authors · 2019-11-12
> **Response part 3**
>
> Comment 7:
>  to further understand the difficulty in SMILES variations, it would be better to show the averaged cardinality of SMILES representation per molecular.
>
> Response 7:
>
> The number of spanning trees for a given molecular graph can be calculated with Kirchoff’s matrix tree theorem. We find the average number of spanning trees in ZINC to be 500. As we train for 5000 epochs, and use 10 SMILES strings per molecule per epoch, we have ample opportunity to explore the space of spanning trees of a molecule if we dynamically make our SMILES strings. Preliminary experiments that we have done show diminishing returns in using more than 10 SMILES strings per molecule with unique spanning trees.
>
>
>
> 1:Jason Tyler Rolfe. Discrete variational autoencoders. arXiv preprint arXiv:1609.02200, 2016.
> 2:Chen Xi, et al. "Variational lossy autoencoder." arXiv preprint arXiv:1611.02731,2016.
> 3:Tomczak, Jakub M., and Max Welling. "VAE with a VampPrior." arXiv preprint arXiv:1705.07120 (2017).
> 4:Kingma, D. P.; Mohamed, S.; Rezende, D. J.; Welling, M. Semi-Supervised Learning with Deep Generative Models. Proceedings of the 28th Annual Conference on Neural Information Processing Systems. 2014; pp 3581–3589.
> 5: Seokho Kang and Kyunghyun Cho. Conditional molecular design with deep generative models. arXiv preprint arXiv:1805.00108, 2018.
> 6:Wengong Jin, Regina Barzilay, and Tommi Jaakkola. Junction tree variational autoencoder for molecular graph generation. arXiv preprint arXiv:1802.04364, 2018.

---

### Official Review · AnonReviewer2 · 2019-10-26
**Official Blind Review #2**

**Rating:** 6

**Review:**

The authors describie a novel variational autoencoder like method for molecules. Instead of using graph neural networks, the authors hava an approach based on SMILES which encode molecules as strings. To avoid the problem that any given molecule may be represented by multiple SMILES strings, the authors consider an encoder that makes use of several random SMILES representations of the input molecule. These are preprocessed using recurrent neural networks generating an average representation by pooling the representations generated with each SMILES sequence for each atom in the input molecule. This reduces the number of operations needed to share information across all the atom in the
molecule (from N^2 in graph neural networks to MN in the proposed approach with M different SMILES representations of the input molecule). The model decodes then into a disjoint set of SMILES strings different from those used at the
input. This enforces the model to learn a bijective mapping between molecules and latent representations. The model trains also jointly a property regressor, linear or logistic. They do constrained optimization in the latent space, satying
within a reparameterized shell with most of the probability mass for the data. The proposed model can also do semi-supervised and supervised prediction tasks.

Clarity:

The paper is very clearly written and it is very easy to read. It contains a very detailed description of previous work.

Quality:

The proposed model is very reasonable and well-motivated. The experiments performed are exhaustive and informative enough to show that the proposed model and algorithms are useful in practice.

Novelty:

The proposed encoder/decoder model based on multiple SMILES representation is novel up to my knowledge.

Significance:

The experiments show that the proposed method can outperform previous ones. However, I miss additional evaluations using existing frameworks such as  Guacamol.


**Experience Assessment:**

I have published one or two papers in this area.

**Review Assessment: Checking Correctness Of Derivations And Theory:**

I assessed the sensibility of the derivations and theory.

**Review Assessment: Checking Correctness Of Experiments:**

I assessed the sensibility of the experiments.

**Review Assessment: Thoroughness In Paper Reading:**

I read the paper at least twice and used my best judgement in assessing the paper.

---

> ### Author Response · Authors · 2019-11-12
> **Response**
>
> Thank you for your encouraging comments and precise description of our work. We agree that more benchmarks like Guacamol and MOSES would further strengthen our results [Brown et al., 2019; Polykovskiy et al., 2018]. We are currently working on this along with other new benchmarks to be disclosed in a future publication.
>
> 1:Brown, N., Fiscato, M., Segler, M. H., & Vaucher, A. C. (2019). Guacamol: benchmarking models for de novo molecular design. Journal of chemical information and modeling, 59(3), 1096-1108.
> 2:Polykovskiy, D., Zhebrak, A., Sanchez-Lengeling, B., Golovanov, S., Tatanov, O., Belyaev, S., ... & Kadurin, A. (2018). Molecular sets (moses): a benchmarking platform for molecular generation models. arXiv preprint arXiv:1811.12823.

---

### Official Review · AnonReviewer3 · 2019-10-27
**Official Blind Review #3**

**Rating:** 3

**Review:**

The authors present a method All SMILES VAE that’s used for predicting chemical properties of small molecules and also for optimizing the structures of these molecules. The authors evaluate their model on the Zinc250K and Tox21 dataset and report that they are able to exceed the previous SOTA. This paper was well written, and did a good job with explanations and illustrations.

The central idea is to use a RNN to learn representations of strings encoding molecular structures (SMILE strings). SMILE strings are constructed by a DFS traversal over molecular graph structures. The authors choose to feed in distinct SMILE encodings of the same molecule in parallel, resulting in a stacked RNN architecture. They observe that since SMILE strings are DFS traversals of the molecular graph, propagation in the RNN corresponds to sequential message passing steps between nodes in the graph. This is in contrast to a graph neural network wherein all nodes simultaneously broadcast messages to their neighbors at every propagation step.

While it’s interesting to be able to optimize molecules in the space of SMILE strings, the impact is less clear. The authors mention in Sec 3.1, that graph models e.g. GCNs have higher overall compute complexity O(b^2), compared to their method which has O(Mb); however this does not do complete justice. Most graph propagation operations benefit from heavily parallelizable sparse matrix operations on GPUs. Infact, they can actually be much faster than RNNs in practice since GCNs need only a fixed number of propagation steps (independent of the number of bonds).

In Sec 3.2, the authors describe their approach for constraining the space of molecular optimization. While this may lead to directed searches, it will prevent truly novel molecules from being synthesized. The authors will need to address and provide experiments with unconstrained search.

It was not clear if the authors implemented any of the baselines? It seems from the text of figure 5, that the SSVAE and GraphConv results have been taken directly from the paper. The authors can strengthen their claim by replicating the results of baselines and making sure that they agree.

Some clarification questions:
 - For optimized/predicted strings, which are presumably novel molecules not in the dataset, how is the true chemical property  e.g. logP, determined?
 - In Figure 6, from steps 30 - 40, it seems that the predicted logP goes up but the true logP stays the same in a few cases. Why is this the case?

Overall, I think that this paper has some interesting ideas and is well written. However, the novelty and impact of the model is somewhat lacking. If this were introducing a new application area to this field, then I think the case for acceptance could have been stronger, however there has already been a lot of work related to molecular property prediction/design.

Also, AC please note, I would have given the paper a score of 5, but Openreview only gave me a choice between 3 and 6. So, I went with 3, but please consider this to be a 5.

**Experience Assessment:**

I have published in this field for several years.

**Review Assessment: Checking Correctness Of Derivations And Theory:**

I assessed the sensibility of the derivations and theory.

**Review Assessment: Checking Correctness Of Experiments:**

I assessed the sensibility of the experiments.

**Review Assessment: Thoroughness In Paper Reading:**

I read the paper at least twice and used my best judgement in assessing the paper.

---

> ### Author Response · Authors · 2019-11-12
> **Response part 1**
>
> Comment 1:
> it’s interesting to be able to optimize molecules in the space of SMILE strings, the impact is less clear.
> Response1:
> Our model is over molecules, rather than individual SMILES strings. The VAE architecture effectively takes a molecular graph as input, and dynamically generates n SMILES strings representing that molecule as the target of the decoder. Simultaneously, the encoder takes the molecular graph as input and simultaneously generates m independents SMILES strings representing the molecule to feed into the subsequent stacks of RNNs. Since the SMILES-encoding of the encoder is independent of the SMILES-encoding of the decoder, the latent representation must capture the molecule as a whole, and decode to all possible SMILES strings representing the molecule. As a result, optimization in the latent space is properly in the space of molecules, rather than SMILES strings. Rather, we use SMILES strings as the initial component of a SMILES-based encoder and the final component of a SMILES-based decoder for molecular graphs, analogous to graph convolutions and iterative graph construction in other molecular graph-based generative models.
>
> We train the model to be able to decode to any SMILES string from a given point in latent space, we expect that point to manifest as an abstract representation of the molecule, as opposed to a specific SMILES. Our optimization is in the space of molecules, rather than SMILES strings.
>
> Comment 2:
> Most graph propagation operations benefit from heavily parallelizable sparse matrix operations on GPUs.
>
> Response 2:
> We agree that sparse matrix operations can be efficiently parallelized on GPUs, and take this into account in our analysis by conservatively assuming that each graph convolutional (GC) layer takes only O(b). To further highlight this fact, we have added a note to the paper that GC operations can be efficiently parallelized with sparse GPU operations, as it is important for practical implementation. We further agree that in practise, a single RNN layer is slower than a single GC layer, although big O does not traditionally differentiate parallel and sequential processes.
>
> However, to produce a nonlinear representation that encompasses an entire molecule, GCs utilizing sparse matrix operations require a number of iterations equal to at least half the graph diameter (which is O(b)), giving us O(b^2). Without a number of iterations equal to at least half the graph diameter, information cannot be transferred from all atoms to any single location in the molecular graph. To transfer information from all atoms to all locations in the molecular graph, a number of iterations equal to the graph diameter is required. In practice, many times this number of iterations will generally be required to flexibly combine information from throughout the molecule. Correspondingly, convolutional networks on images use hundreds of layers; moreover, they reduce the resolution with pooling layers to sizes as small as 7x7, so a few layers are sufficient to integrate information throughout the image. While many previous works have used a fixed number of graph convolution layers (generally 3-7), the complexity of GCs is O(b^2) if information must be passed across the entire graph.
>
> While it is possible to lower-bound the graph diameter in terms of the number of bonds, this bound is only realized in densely connected graphs, whereas bonds amongst atoms in molecules are sparse and often planar. As a result, the graph diameter can be large, even when the number of bonds is modest. In the ZINC250k dataset, the mean graph diameter is 11.1; the maximum is 24. Typical implementations of graph convolution use only 3 to 7 rounds of message passing [Duvenaud et al., 2015; Gilmeret al., 2017; Jin et al., 2018; Kearnes et al., 2016; Liu et al., 2018; Samanta et al., 2018; You et al., 2018], and so cannot propagate information across most molecules in this dataset.
>
> In contrast, an RNN on a SMILES string of the molecule, in which some bound atoms are processed sequentially, while other bound atoms are far apart in the string, will pass information across the whole string in a single layer. Our novel use of multiple SMILES strings allows information to pass quickly through long paths in the molecule, while efficiently representing all paths (using multiple SMILES strings).

---

> ### Author Response · Authors · 2019-11-12
> **Response part 2**
>
> Comment 3:
> In Sec 3.2, the authors describe their approach for constraining the space of molecular optimization. While this may lead to directed searches, it will prevent truly novel molecules from being synthesized. The authors will need to address and provide experiments with unconstrained search.
>
>
> Response 3:
> The Gaussian Annulus Theorem [Blum et al., 2017] shows that nearly all the probability of an n dimensional spherical Gaussian with unit variance is concentrated in a thin annulus of width O(1) at radius sqrt(n). After subtracting the mean and dividing out the standard deviation, each reparameterized layer of our hierarchy is a unit variance spherical Gaussian with 100 dimensions (n=100), so 90% of the probability mass lies between radius 7.4 and 12.6 (sqrt(n) = 10). The sphere with radius 10 lies within 2.6 of almost all the probability mass of the prior.
>
> In Section 4, we sample 50,000 molecules from the prior, and find that 99.958% are novel relative to the training set (all are unique). Intuitively, there is no reason to believe that samples constrained to the sphere, with radius exactly 10 rather than approximately 10, would behave differently. Moreover, optimization within the sphere will naturally drive towards regions of the latent space corresponding to molecules with properties more extreme than any in the training set, which must necessarily be novel. Indeed, none of the optimized molecules depicted in Figure 7 are present in the training dataset. The maximum penalized logP in the training dataset is about 5; considerably less than the penalized logPs of the molecules in Figure 7(a), all of which are greater than this.
>
> To further empirically substantiate this intuition, we have provided experiments with unconstrained search in Table 7 of Section C.4 in the supplementary materials, where we show that the radius constraint improves the efficacy of property optimization by about two times, and allows us to find molecules with property values much greater than those found in the training set.
>
> Finally, to provide further direct evidence that we can find novel molecules not found in the training set with the constraint. We have done an additional experiment, where we sample from the prior, and project onto the radius. We then look at the novelty of the molecules found on on the constraint relative to the training set, and the uniqueness (no molecules are repeated). These benchmarks are found in Guacamol and Moses.
> We find:
>
> Novelty: 100%
> Uniqueness: 100%
>
> Thus our radius constraint finds unique molecules.
>
> Comment 4:
> It was not clear if the authors implemented any of the baselines? It seems from the text of figure 5, that the SSVAE and GraphConv results have been taken directly from the paper.
>
> Response 4:
>
> As is standard in the molecular optimization literature [Gomez-Bombarelli et al., 2018; Kang and Cho, 2018; Zhou et al., 2018], we compute logP using Crippen’s method, as implemented in RDKit [Wildman and Crippen, 1999 ; Landrum, 2006]. This approach is universal in the molecular optimization literature, since wet-lab measurements of logP cannot be efficiently or reproducibly produced for novel molecules.
>
> We optimize novel molecules using only the predicted logP from our regressors, and then calculate the ‘true’ logP using RDkit. We only use the molecules with the top 100 predicted logPs to choose those for which we check the ground truth logP.
>
> Comment 5:
> - In Figure 6, from steps 30 - 40, it seems that the predicted logP goes up but the true logP stays the same in a few cases. Why is this the case?
>
> Response 5:
> In Figure 6, predicted logP becomes less accurate outside of the support of the training set. The ZINC250k dataset includes molecules with penalized logPs (the log octanol-water partition coefficient minus the synthetic accessibility score and the number of rings with more than six atoms) ranging from -13 to 5. Correspondingly, in Figure 6, the predicted penalized logP becomes a noisier estimator of the true penalized logP once the optimization progresses beyond a penalized logP of 5.
>
> Comment 6:
> However, the novelty and impact of the model is somewhat lacking. If this were introducing a new application area to this field, then I think the case for acceptance could have been stronger, however there has already been a lot of work related to molecular property prediction/design.
>
> Response 6:
> Our novel contributions are primarily architectural. In addition to the use of multiple parallel RNN stacks processing multiple SMILES representations of a single molecule, we introduce a novel information exchange mechanism amongst these parallel representations, using an attentional mechanism to harmonize the atom encodings across the representations. We also introduce the Gaussian annulus constraint in the optimization. Using ablation experiments, we show that these novel contributions all contribute to improve the performance of our algorithm beyond the previous state-of-the-art.

---

> ### Author Response · Authors · 2019-11-12
> **Citations From Responses**
>
> 1: Avrim Blum, John Hopcroft, and Ravi Kannan. Foundations of Data Science. June 2017. URL https://www. microsoft.com/en-us/research/publication/foundations-of-data-science-2/.
> 2: Seokho Kang and Kyunghyun Cho. Conditional molecular design with deep generative models. arXiv preprint arXiv:1805.00108, 2018.
> 3:Rafael Go ́mez-Bombarelli, Jennifer N Wei, David Duvenaud, Jose ́ Miguel Herna ́ndez-Lobato, Benjam ́ın Sa ́nchez- Lengeling, Dennis Sheberla, Jorge Aguilera-Iparraguirre, Timothy D Hirzel, Ryan P Adams, and Ala ́n Aspuru-Guzik. Automatic chemical design using a data-driven continuous representation of molecules. ACS central science, 4(2): 268–276, 2018.
> 4:Zhenpeng Zhou, Steven Kearnes, Li Li, Richard N Zare, and Patrick Riley. Optimization of molecules via deep reinforcement learning. arXiv preprint arXiv:1810.08678, 2018.
> 5:Wildman, Scott A., and Gordon M. Crippen. Prediction of physicochemical parameters by atomic contributions. Journal of chemical information and computer sciences, 868-873, 1999.
> 6:Greg Landrum et al. Rdkit: Open-source cheminformatics, 2006.
> 7:David K Duvenaud, Dougal Maclaurin, Jorge Iparraguirre, Rafael Bombarell, Timothy Hirzel, Alan Aspuru-Guzik, and Ryan P Adams. Convolutional networks on graphs for learning molecular fingerprints. In Advances in neural information processing systems, pp. 2224–2232, 2015.
> 8:Justin Gilmer, Samuel S Schoenholz, Patrick F Riley, Oriol Vinyals, and George E Dahl. Neural message passing for quantum chemistry. In Proceedings of the 34th International Conference on Machine Learning-Volume 70, pp. 1263–1272. JMLR. org, 2017.
> 9:Wengong Jin, Regina Barzilay, and Tommi Jaakkola. Junction tree variational autoencoder for molecular graph generation. arXiv preprint arXiv:1802.04364, 2018.
> 10:Steven Kearnes, Kevin McCloskey, Marc Berndl, Vijay Pande, and Patrick Riley. Molecular graph convolutions: moving beyond fingerprints. Journal of computer-aided molecular design, 30(8):595–608, 2016.
> 11:Qi Liu, Miltiadis Allamanis, Marc Brockschmidt, and Alexander L Gaunt. Constrained graph variational autoencoders for molecule design. arXiv preprint arXiv:1805.09076, 2018.
> 12:Bidisha Samanta, Abir De, Gourhari Jana, Pratim Kumar Chattaraj, Niloy Ganguly, and Manuel Gomez-Rodriguez. NeVAE: a deep generative model for molecular graphs. arXiv preprint arXiv:1802.05283, 2018.
> 13:Jiaxuan You, Bowen Liu, Rex Ying, Vijay Pande, and Jure Leskovec. Graph convolutional policy network for goal-directed molecular graph generation. arXiv preprint arXiv:1806.02473, 2018.

---

### Public Comment · ~AkshatKumar_Nigam1 · 2019-10-04
**Normalization constants**

Dear Authors,

We would like to begin by saying that we thoroughly enjoyed reading your paper!

However, we would like to call your attention to a potential oversight in Table 2 and the corresponding molecules in Figure 7a.

The penalized logP scores of the three molecules of Figure 7a are reported as (16.42, 16.32, 16.21) in Table 2. We were checking the penalized logP scores of these molecules and we suspect that you report non-normalized values. However, all literature values (i.e. JT-VAE, etc.) are normalized scores with normalization constants based on the ZINC dataset.
The mean and standard deviation for the zinc dataset are:
logP       : 2.4729421499641497   &  1.4157879815362406
SAS        : 3.0470797085649894   &  0.830643172314514
RingPenalty: 0.038131530820234766 &  0.2240274735210179
If the above constants are used, the score for the molecules become: (12.48, 12.19, 12.30).

Please let us know if the above raises any questions.
Regards,
AkshatKumar Nigam, Pascal Friederich, Mario Krenn, Alán Aspuru-Guzik

---

> ### Author Response · Authors · 2019-10-05
> **Fixed normalization issue**
>
> Dear AkshatKumar Nigam, Pascal Friederich, Mario Krenn, and Alán Aspuru-Guzik,
>
> Thank you for your helpful comment on our ICLR submission! We offer our sincere apologies for missing the normalization implicit in the reporting of penalized logP. Reference to this normalization appears in the supplementary materials of the Junction Tree VAE (Jin et al.,2018). It seems to go unmentioned in the text, but is present in the code, of the Grammar VAE (Kusner et al., 2017) and Graph Convolutional Policy Network (You et al., 2018). So far as we can tell, Molecule Deep Q-Networks (MolDQN; Zhou et al, 2019) does not appear to normalize penalized logP. We recomputed the unnormalized penalized logP for their best molecules, and it matches their reported values.
>
> We have updated our penalized logP values to include normalization. With KL unscaled, we obtain 29.80, 29.76, 29.11. With the KL scaled, we obtain 12.31, 12.13, 12.01. We have also recomputed the normalized penalized logP for the best molecules found by MolDQN, for which we obtain 8.93, 8.93, 8.91. Normalization of penalized logP does not substantially change the relative improvement of our result over MolDQN, the rankings of all algorithms remains unchanged, and our results remain state-of-the-art. We will upload the corrected copy as soon as possible.
>
> Regards,
>
> The authors of 'All SMILES Variational Autoencoder for Molecular Property Prediction and Optimization'

---

### Decision · Program_Chairs · 2019-12-19

**Decision:**

Reject

**Comment:**

The paper proposes All SMILES VAE which can capture the chemical properties of small molecules and also optimize the structures of these molecules. The model achieves significantly performance improvement over existing methods on the Zinc250K and Tox21 datasets.

Overall it is a very solid paper - it addresses an important problem, provides detailed description of the proposed method and shows promising experiment results. The work could be a landmark piece, leading to major impacts in the field. However, given its potential,  the paper could benefit from major revisions of the draft. Below are some suggestions on improving the work:
1. The current version contains a lot of materials. It tries to strike the balance between machine learning methodology and details of the application domain. But the reality is that the lack of architecture details and some sloppy definitions of ML terms make it hard for readers to fully appreciate the methodology novelty.

2. There is still room for improvement in experiments. As suggested in the review, more datasets should be used to evaluate the proposed model. Since it is hard to provide theoretic analysis of the proposed model,  extensive experiments should be provided.

3. The complexity analysis is not fully convincing. Some fair comparison with the alternative approaches should be provided.

In summary, it is a paper with big potentials. The current version is a step away from being ready for publication. We hope the reviews can help improve the paper for a strong publication in the future.